# Feed-forward recruitment of electrical synapses enhances synchronous spiking in the mouse cerebellar cortex

Andreas Hoehne[1,2], Maureen H McFadden[1], David A DiGregorio[1]*

[1]Laboratory of Synapse and Circuit Dynamics, Institut Pasteur, Paris Cedex, France; [2]Sorbonne University, ED3C, Paris, France

**Abstract** In the cerebellar cortex, molecular layer interneurons use chemical and electrical synapses to form subnetworks that fine-tune the spiking output of the cerebellum. Although electrical synapses can entrain activity within neuronal assemblies, their role in feed-forward circuits is less well explored. By combining whole-cell patch-clamp and 2-photon laser scanning microscopy of basket cells (BCs), we found that classical excitatory postsynaptic currents (EPSCs) are followed by $GABA_A$ receptor-independent outward currents, reflecting the hyperpolarization component of spikelets (a synapse-evoked action potential passively propagating from electrically coupled neighbors). FF recruitment of the spikelet-mediated inhibition curtails the integration time window of concomitant excitatory postsynaptic potentials (EPSPs) and dampens their temporal integration. In contrast with GABAergic-mediated feed-forward inhibition, the depolarizing component of spikelets transiently increases the peak amplitude of EPSPs, and thus postsynaptic spiking probability. Therefore, spikelet transmission can propagate within the BC network to generate synchronous inhibition of Purkinje cells, which can entrain cerebellar output for driving temporally precise behaviors.

*For correspondence:
david.digregorio@pasteur.fr

Competing interests: The authors declare that no competing interests exist.

## Introduction

GABAergic interneurons play important but diverse roles in gating, routing and modulating excitatory information flow within neural circuits through inhibitory chemical synapses. These different computational functions are classified according to canonical wiring motifs (*Feldmeyer et al., 2018*; *Isaacson and Scanziani, 2011*): feed-back inhibition supports rate-based modulation of excitation through gain modulation and increased dynamic range of the levels of excitatory input that can be encoded within a circuit, while feed-forward (FF) inhibition permits precise regulation of postsynaptic spike-timing by sharpening excitatory postsynaptic potential (EPSP)-spike coupling (*Blot et al., 2016*; *Mittmann et al., 2005*; *Pouille and Scanziani, 2001*). FF motifs can also expand the dynamic range of inputs which can be represented by principal cells (*Pouille et al., 2009*). However, how FF inhibitory motifs can be used to regulate interneuron firing is less well studied.

Interneurons communicate with each other through both chemical and electrical synapses. While chemical synapses can regulate interneuron firing in the same way as for principal neurons (*Mittmann et al., 2005*), the presence of gap junctions that couple neurons directly as resistive elements are thought to generate emergent properties such as oscillations and synchronized firing within neuronal assemblies (*Beierlein et al., 2000*; *Bennett and Zukin, 2004*; *Draguhn et al., 1998*; *Gutierrez et al., 2013*; *Maex and De Schutter, 2007*; *Ostojic et al., 2009*; *van Welie et al., 2016*). Electrical synapses between molecular layer interneurons (MLIs) within the cerebellum have been shown to mediate synchronized firing (*Gaffield and Christie, 2017*; *Mann-Metzer and Yarom, 1999*) and sequence detection (*Alcami, 2018*), while those between Golgi cells have been shown to either synchronize or desynchronize cerebellar cortical

network activity (*Dugué et al., 2009*; *van Welie et al., 2016*; *Vervaeke et al., 2010*). Finally, the presence of electrical synapses has been shown to modulate passive properties in interneurons (*Alcami, 2018*; *Amsalem et al., 2016*; *Hjorth et al., 2009*). However, despite theoretical studies showing the influence of electrical synapses in information processing within FF circuits (*Pham and Haas, 2019*), experimental evidence that electrical synapses modify temporally coded information within FF neural circuits has not been demonstrated.

Owing to the low-pass filtering properties of the cell membrane, presynaptic APs are heavily filtered and consequently detected as 'spikelets' in electrically coupled postsynaptic cells. The relative contribution of the depolarizing component (*Hu and Agmon, 2015*; *Mann-Metzer and Yarom, 1999*) or the after-hyperpolarization component (*Dugué et al., 2009*; *Vervaeke et al., 2010*) of the presynaptic AP to the spikelet waveform can vary across cell types and determines if electrical synapses mediate net excitation or inhibition. Alterations in the spikelet waveform can arise from alterations in presynaptic AP shape, which can be modulated by variations in the membrane potential of unperturbed neurons when they are not spiking (resting membrane potential) (*Otsuka and Kawaguchi, 2013*; *Russo et al., 2013*). Theoretical studies show that net depolarizing spikelets will drive synchronization within neuronal networks, while hyperpolarizing spikelets will generate bistable networks which either oscillate synchronously or remain in asynchronous states (*Ostojic et al., 2009*). Thus, the contribution of electrical synapses to network computations depends on the precise polarity of the spikelet.

In the cerebellar cortex, MLIs are known to mediate feed-forward inhibition (FFI) onto each other and Purkinje cells (PCs) by the means of GABAergic synapses. Coherent firing of MLIs is required for encoding precise movement kinematics (*Gaffield and Christie, 2017*). Specifically, basket cells (BCs) have been shown to precisely control the timing of PC activity with millisecond precision (*Arlt and Häusser, 2020*; *Blot et al., 2016*). MLIs in young animals have also been shown to be electrically coupled via the gap junction protein Cx36, likely located in the dendrites (*Alcami and Marty, 2013*). It has been shown that there is an increasing spatial gradient of occurrence and strength of electrical connectivity between MLIs from increasing depths in molecular layer (*Rieubland et al., 2014*). Since MLIs located in the deepest (inner) third of the molecular layer are likely to be BCs (*Sultan and Bower, 1998*), we investigated if electrical transmission could be recruited in a feed-forward (FF) manner in the BC population, and how it modulates EPSP-spike coupling.

We found that electrical stimulation of parallel fiber (PF) excitatory inputs reliably elicited excitatory postsynaptic currents (EPSCs), followed rapidly by an outward current that was sensitive to gap junction antagonists and Cx36 knockdown, consistent with suprathreshold, PF-mediated recruitment of an electrically coupled neuron delivering a spikelet to its neighbor. Paired whole-cell recordings from electrically coupled BCs showed that the spikelets were predominantly inhibitory due to the depolarized presynaptic resting potential. Consequently, the synaptic recruitment of an electrically coupled MLI neighbor displays some of the characteristic features of FFI: temporal shortening of single EPSPs and dampening of temporal summation. Unlike chemical FFI, however, we demonstrate that EPSP-evoked spikelets can amplify synchronous compound synaptic responses, and consequently increase the probability of AP firing over a brief time window. Thus, synapse-evoked spikelet recruitment in a FF motif can act as a mechanism for temporal contrast enhancement of information flow within neural circuits.

## Results

### PF-triggered outward currents are mediated by electrical synapses in cerebellar BCs

MLIs found in the inner third of the molecular layer of the cerebellar cortex are more likely to be BCs, defined by their basket-like axonal projections targeting PC somata and axon initial segment (*Sultan and Bower, 1998*). Because MLIs from the inner-third of the molecular layer have been shown to more frequently form electrical synapses (*Rieubland et al., 2014*), we examined whether electrical synapses could be recruited in a FF manner between electrically coupled BCs. Two-photon laser scanning microscopy (2PLSM) of whole-cell patch-clamped inner MLIs revealed that more than 95% of cells with basket-like axon collaterals also presented a dendritic tree that traversed nearly the entire molecular layer (174 ± 4.9 µm, n = 30 cells - see *Figure 1A* for a representative example).

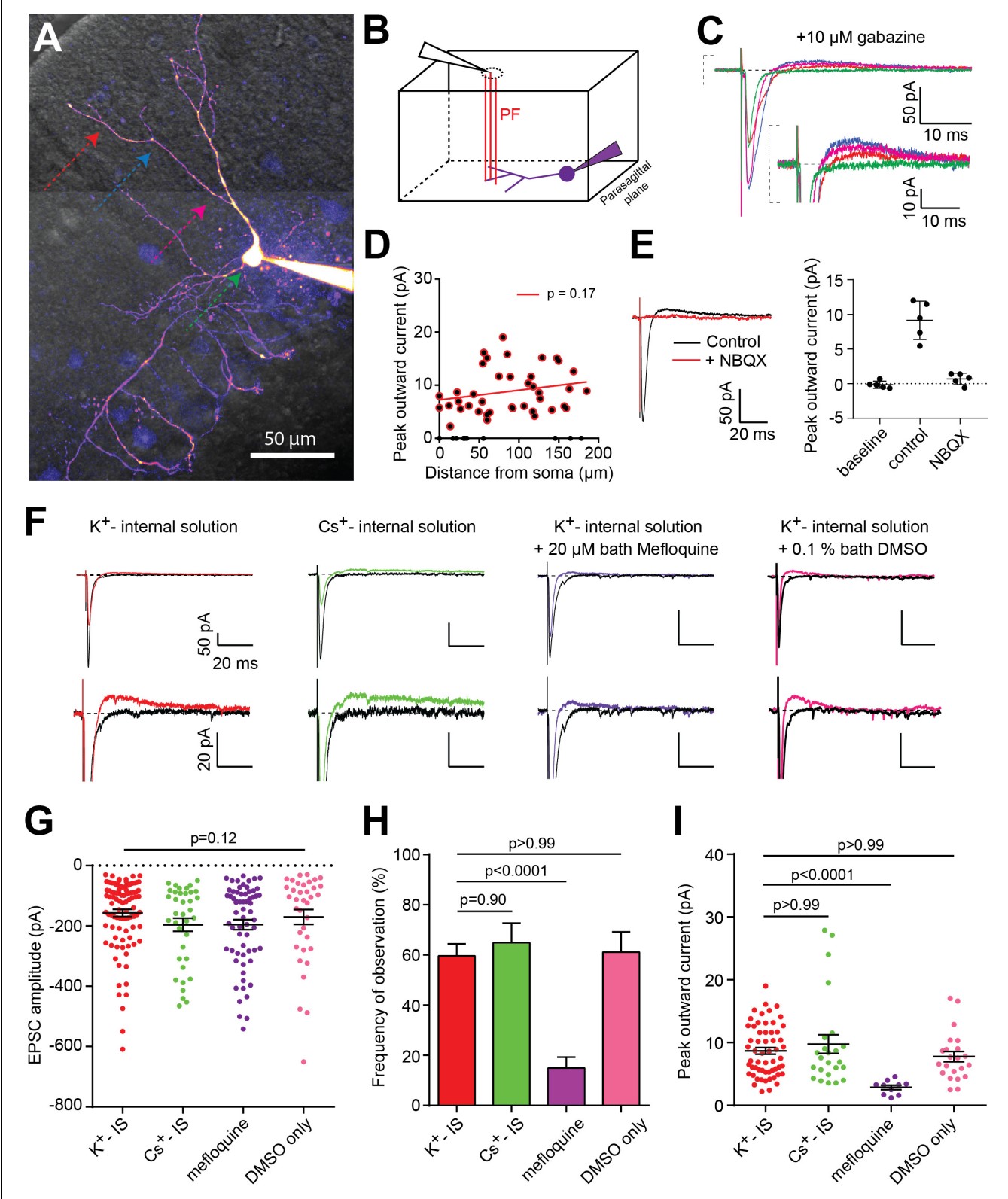

**Figure 1.** PF-basket cell (BC) synapse-evoked outward currents that are insensititive to GABAergic receptor antagonists are sensitive to gap junction antagonists. (**A**) 2-photon laser scanning microscopy image (2PLSM - maximal intensity projection) of a cerebellar BC patch-loaded with 20 μM Alexa-594. Color arrows indicate the positions of the stimulating pipette along the somato-dendritic axis. (**B**) Schematic diagram showing the stimulation pipette (black triangle) and granule cell axons or parallel fibers (PFs - red lines) projecting perpendicularly to, and synapsing onto, the dendritic tree of

*Figure 1 continued on next page*

**Figure 1 continued**

the patch-clamped BC (purple) in a parasagittal cerebellar slice configuration. (C) Averaged excitatory postsynaptic currents (EPSCs) were recorded from the cell in A and in the presence of 10 µM gabazine, following a single brief (50 µs) extracellular voltage stimulation of a PF beam using a monopolar glass electrode placed at different locations in the dendritic tree (same color code as in A). (D) Average peak amplitude of the outward current versus distance between synaptic current entry and somatic compartment. Linear regression analysis reveals a non-significant relationship between the two parameters (p=0.17, n = 52 stimulation sites over 13 cells; n = 40 stimulation sites with significant outward currents). (E) Representative experiment showing 10 µM NBQX block of both inward and outward currents. Right, summary results from 5 cells. Error bars are standard deviation. (F) Representative traces of PF-mediated synaptic responses recorded in four different pharmacological conditions. Two traces recorded when the stimulus electrode was placed at a site with (color) or without synapse-evoked outward currents (black). (G) EPSC amplitude following PF stimulation is not significantly different between the four groups (Kruskal-Wallis test, p-value=0.12 K$^+$-IS: n = 96 stimulation sites over 26 cells; Cs$^+$-IS: n = 37 sites over 10 cells; Mefloquine: n = 64 sites over 16 cells; DMSO only: n = 36 sites over 11 cells). (H) The frequency of observing an outward current is significantly smaller only in the mefloquine-treated group (Brown-Forsythe, F = 57.13, p-value<0.0001; followed by Dunnett's multiple comparisons tests; K$^+$-IS: n = 104 stimulation sites; Cs$^+$-IS: n = 37 sites; Mefloquine: n = 67 sites; DMSO only: n = 36 sites). Error bars are SEM and calculated assuming binomial statistics (see Materials and methods). (I) Outward current peak amplitude (in cases where it can be detected) is significantly smaller only in the mefloquine-treated group (Kruskal-Wallis test followed by Dunn's multiple comparison tests, p<0.0001; K$^+$-IS: n = 58 stimulation sites over 20 cells; Cs$^+$-IS: n = 24 sites over 10 cells; Mefloquine: n = 10 sites over 8 cells; DMSO only: n = 22 sites over 10 cells). See also *Figure 1—source data 1*.

The online version of this article includes the following source data and figure supplement(s) for figure 1:

**Source data 1.** PF-basket cell (BC) synapse-evoked outward currents that are insensititive to GABAergic receptor antagonists are sensitive to gap junction antagonists.

**Figure supplement 1.** The amplitude of direct excitatory postsynaptic currents (EPSCs) and outward currents are not correlated.

**Figure supplement 1—source data 1.** The amplitude of direct EPSCs and outward currents are not correlated.

All subsequent recordings were made from inner MLIs with dendritic or axonal morphologies characteristic of BCs (*Sultan and Bower, 1998*).

Simultaneous Dodt contrast imaging and 2PLSM were used to target extracellular stimulation of PFs at different locations along the dendritic tree (*Figure 1A–C*). Stimulation intensity was adjusted to obtain stable EPSCs and 50 Hz PPR (generally 10–15 V above threshold; *Abrahamsson et al., 2012*) in the presence of saturating concentrations of the GABA$_A$ receptor blocker, gabazine (10 µM [*Ueno et al., 1997*]). Despite blocking inhibition, we often observed a fast EPSC component and a gabazine-insensitive outward component when stimulating PF axons contacting BC dendrites (*Figure 1*). This outward current was detected in approximately 60% of stimulation locations (*Figure 1H*, red bar) across 26 cells, twenty-one of which displayed significant outward currents in at least one stimulation site (see Materials and methods). Outward currents were also detected all along the dendritic tree, with an amplitude independent of the distance from stimulation site to the soma (*Figure 1D*). Bath application of 10 µM of the AMPA receptor (AMPAR) antagonist, NBQX, eliminated both the EPSC and the subsequent outward current to below detectable levels (*Figure 1E*, n = 5/5 experiments). In the absence of NBQX, increasing the stimulation voltage increased the amplitude of both the inward and outward currents, but single-sweep analysis within each stimulus condition did not reveal any correlation between their peak amplitudes (*Figure 1—figure supplement 1*), despite quantal variability in the peak EPSC. This finding is not consistent with voltage-dependent potassium conductance recruitment by unclamped synaptic currents in dendrites for larger synaptic currents (*Tran-Van-Minh et al., 2016*). To test this more directly, we replaced potassium by cesium in the patch pipette solution to block potassium conductances. We observed no difference in the frequency or amplitude of outward currents (*Figure 1F–I*). We next reasoned that PF stimulation could trigger an AP in electrically-connected neurons, generating a spikelet with a predominant hyperpolarizing (outward) current component following the initial EPSC. Consistent with this hypothesis, 20 µM of mefloquine, a potent blocker of gap junctions (~70% block at 20 µM [*Cruikshank et al., 2004*]) reduced the frequency of observing an outward current to 14.9 ± 4.4%, with those detected currents being 3-fold smaller (8.7 ± 0.5 vs. 2.9 ± 0.3 pA; *Figure 1I*). In conditions where only 0.1% DMSO was added to the artificial cerebrospinal fluid (ACSF), neither the frequency nor the amplitude of the outward current differed from control (*Figure 1H and I*, pink group).

We also examined the frequency of observing PF-mediated outward currents in the absence of Cx36 expression (n = 4 mice; *Figure 2*). In brain slices prepared from mature Cx36 KO mice, we observed a 45% reduction in the frequency of outward currents, and the amplitude of the remaining events was 3-fold smaller than age-matched recordings (4.2 ± 0.8 vs. 12.7 ± 1.8 pA). The remaining

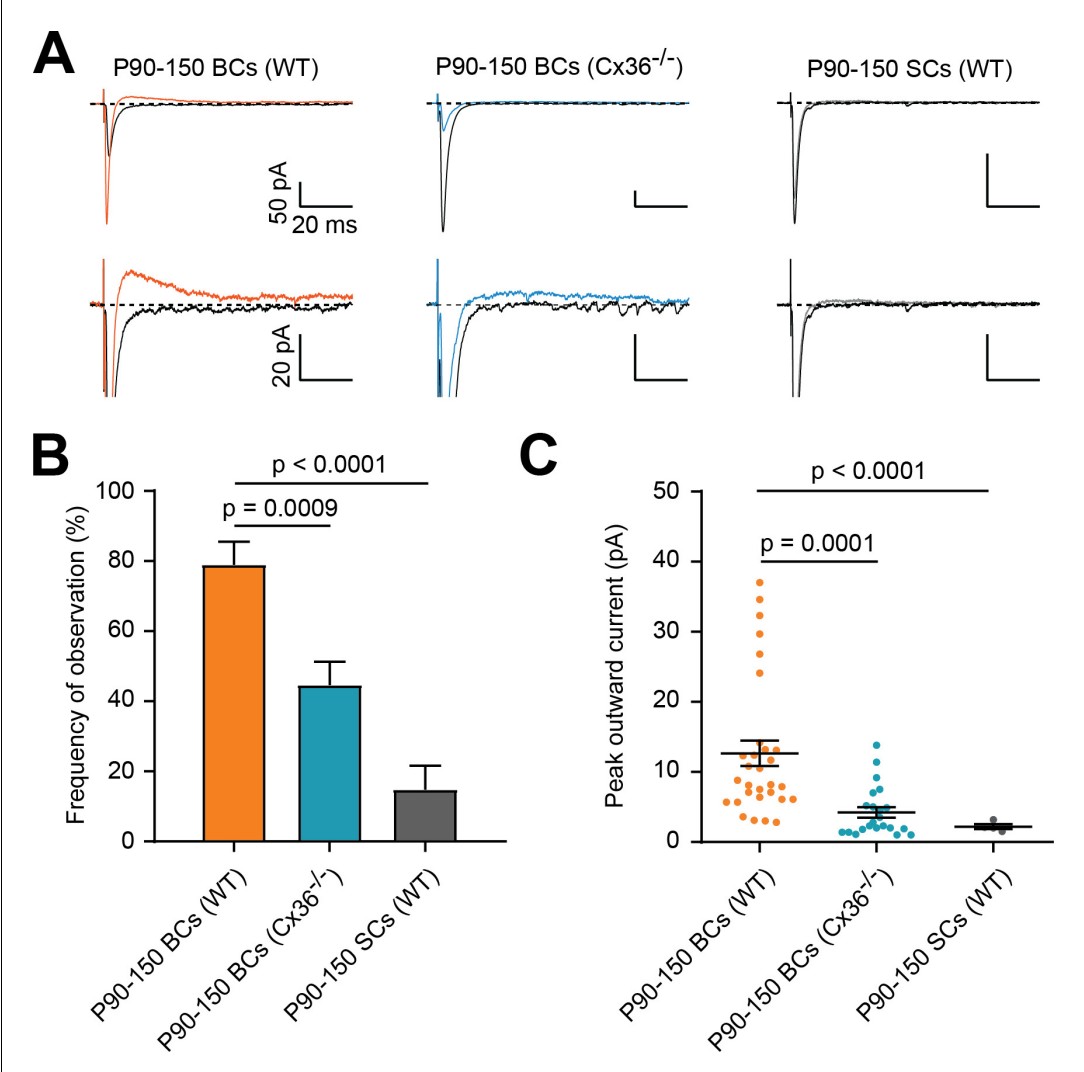

**Figure 2.** Outward currents are reduced in Cx36KO mice and stellate cells (SCs). (**A**) Representative traces of PF-mediated synaptic responses recorded in three different conditions (WT P90-150 BCs, P90-150 BCs in Cx36KO mice, and WT P90-150 SCs). Black and colored trace for each group are averages of 20–30 sweeps at a location without and with a detectable outward current, respectively. Bottom row is an expanded amplitude scale to better visualize outward current. (**B**) The frequency of observing the outward current is significantly smaller in P90-P150 SCs (WT) and BCs (Cx-36 KO), as compared to P90-P150 BCs (WT) (Brown-Forsythe with Dunnetts multiple comparisons test, F = 17.09, p<0.0001; P90-P150 BCs (WT): n = 38 stimulation sites over 11 cells, P90-P150 BCs (Cx36$^{-/-}$): n = 56 sites over 14 cells, P90-P150 SCs (WT): n = 27 sites over 8 cells). (**C**) Outward current peak amplitude is significantly smaller in P90-P150 SCs (WT) and BCs (Cx-36 KO), as compared to P90-P150 BCs (WT) (Kruskal-Wallis test with Dunn's multiple comparisons test, p<0.0001; P90-P150 BCs (WT): n = 30 sites over 10 cells, P90-P150 BCs (Cx36$^{-/-}$): n = 22 sites over 8 cells, P90-P150 SCs (WT): n = 4 sites over 2 cells). See also *Figure 2—source data 1*.

The online version of this article includes the following source data for figure 2:

**Source data 1.** Outward currents are reduced in Cx36KO mice andstellate cells(SCs).

electrical coupling could be due to a late-onset (3–5 months) partial compensatory mechanism, in contrast to previous reports that showed complete block at P10-13 (*Alcami and Marty, 2013*). We also assessed electrical coupling in BC-BC paired whole-cell recordings in Cx36 KO mice, and found that one out of five recordings exhibited bidirectional spikelets and a coupling coefficient (CC) greater than 2% (see below for criteria). This is less than the nearly 60% of BC-BC pairs showing detectable electrical coupling (see below, Figure 6). These alterations in electrical coupling, either using acute block of gap junctions or via Cx36 knockdown, point to gap junctions as the mechanism for generating synapse-evoked outward currents.

We also found that SCs from older WT animals (P90-150) are 4-fold less likely to exhibit such out-ward currents (15 vs. 79%, n = 27 and 38 stimulation sites, respectively) and exhibit 5-fold smaller outward currents (2.2 ± 0.4 vs. 12.7 ± 1.8 pA), consistent with the lower electrical coupling probabili-ties and amplitudes observed previously in P18-23 rats (*Rieubland et al., 2014*). Together, these observations in adult BCs are consistent with the proposal that PF synapse-evoked outward currents in BCs are mediated by electrical synapses formed by gap junctions, and well-poised to mediate a GABAergic-independent FFI (*Mittmann et al., 2005*; *Pouille and Scanziani, 2001*). All subsequent experiments were therefore performed on BCs.

## PF-evoked and direct recruitment of MLI spikelets

Spikelet transmission through electrical synapses has been shown to exhibit a long-lasting inhibi-tory current (*Alcami and Marty, 2013*; *Mann-Metzer and Yarom, 1999*; *Rieubland et al., 2014*). We, therefore, hypothesized that the outward current reflects the after-hyperpolarization of spikelets, (*i.e.*, filtered presynaptic APs) arising from electrically coupled neighbors, which are generated by suprathreshold PF-evoked EPSPs. If true, then extracellular stimulation of a PF beam outside the dendritic tree of the recorded BC, in the presence of gabazine, would depo-larize a coupled neighbor to threshold (*Figure 3A and B*; off-beam stimulation), thereby generat-ing a spikelet-only waveform without the direct EPSC, as described above. Indeed, rapid and small inward currents were invariably observed prior to large relative outward currents (see example cell in *Figure 3A–E*; n = 11/11 stimulation locations from 9 cells). Both current compo-nents increased with increasing extracellular stimulus intensity (from 20 to 50 V), consistent with recruitment of at least one electrically coupled MLI (*Figure 3E*). We also note that this manipula-tion sometimes allowed the isolation of putative subthreshold EPSCs propagating from the pre-synaptic electrical neighbor at intermediate stimulus intensities (in 2/11 stimulation locations; see example in *Figure 3—figure supplement 1*), indicating that spikelet-like responses recorded in this configuration could be a mixture of pure spikelets and EPSCs filtered across electrical synapses.

In order to ensure that spikelet responses were not mixed with putative subthreshold EPSCs, we also performed direct stimulation of neighboring MLIs, in the presence of 10 µM NBQX, to block AMPAR-mediated EPSCs (*Figure 3F and G*). *Figure 3H* shows a representative experiment in which we examined evoked currents at different extracellular stimulus intensities. For a 20 V stimulus, no postsynaptic response was observed, but at 30 and 40 V, significant inward and outward currents were observed in single sweeps (>2 x SD of background, gray region). However, this was only in a fraction of trials. For 50 V stimuli, single-sweep analysis revealed that all responses were comprised of detectable inward and outward currents (p-value<0.05 for two-by-two comparisons of inward and outward currents; *Figure 3H and J*). When looking at the mean of all responses for each stimulation intensity, we found that each time an outward current was detected, it was preceded by a detect-able inward current (n = 17/17 stimulation sites from 13 cells). The increasing size of both the inward and outward synapse-evoked components were presumably due to the recruitment of additional electrically coupled cells. Taken together, these experiments strongly suggest that detected outward currents were systematically preceded by an inward current, which is the electrophysiological signa-ture of spikelet signaling in MLIs, and provide further evidence that the inhibitory currents described in *Figures 1* and *2* arise from the after-hyperpolarization associated with FF recruitment of spikelet transmission.

## Correlation between spikelet amplitude and electrical coupling coefficient

In order to confirm that extracellular stimulation indeed evoked a presynaptic AP which could then propagate across electrical synapses formed by gap junctions to produce a spikelet in the postsynaptic cell, we performed paired BC-BC recordings. Hyperpolarizing current pulses were used to confirm the presence and the strength of coupling coefficients (CCs; see Materials and methods). Spikelet transmission was isolated by blocking GABAergic transmission with gabazine, which did not affect CCs (*Figure 4B and C*). Evoking an action potential in one BC (transmitting) and recording currents in a receiving BC revealed a spikelet with an inward and outward compo-nent at nearly the same moment, with larger inward and outward currents for larger CCs

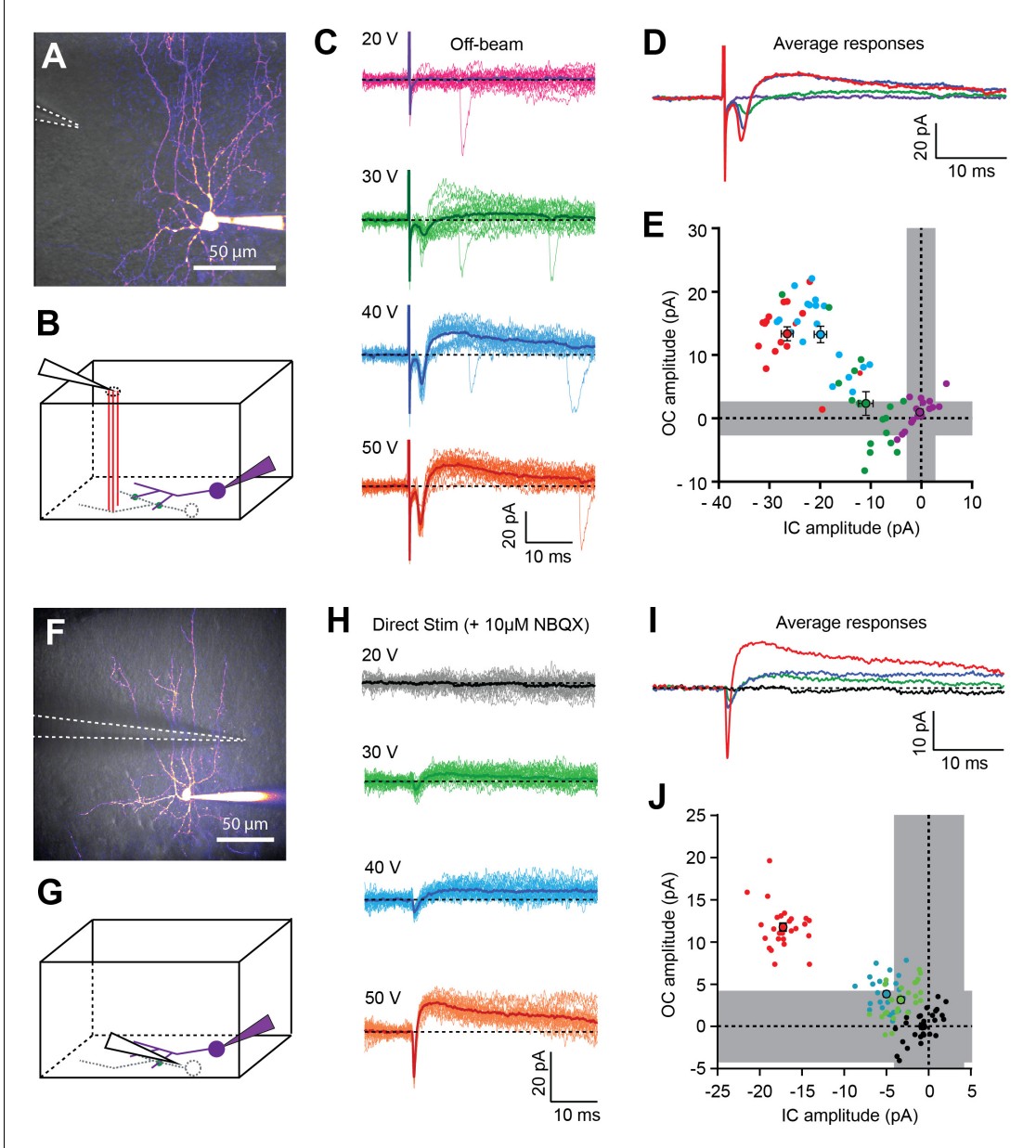

**Figure 3.** In the absence of direct excitatory transmission from PFs, GABAergic-independent outward currents are preceded by a short-lasting inward current. (**A**) 2PLSM image (maximal intensity projection) of a BC loaded with 20 µM Alexa-594. Dashed white lines indicate the position of the extracellular stimulation electrode on the surface of the slice (off-beam stimulation). (**B**) Schematic diagram of a parasagittal slice indicating off-beam stimulation of parallel fibers (PFs - red lines) synapsing onto the dendritic tree of an unpatched neighboring MLI (dashed gray lines), which forms putative electrical synapse(s) (green dots) with the patched cell (purple). (**C**) Postsynaptic current responses recorded in the patched cell in response to a single 50 µs voltage pulse at different stimulation intensities (20–50V). Dark lines represent averages of 15–30 single trials. (**D**) Superimposition of average responses from C, shown for visual comparison of the differences in mean peak amplitude of inward and outward currents. (**E**) Summary plot of peak amplitudes of inward and outward currents from individual trials shown in C, with corresponding averages +/- SEM represented by larger dots. gray regions represent 2*SD of baseline values (averaged over all traces). (**F-J**) Similar as in A-E, but spikelets were elicited by direct extracellular stimulation of a neighboring putative BC in the presence of 10 µM NBQX in the bathing solution to block AMPAR-mediated EPSCs. See also *Figure 3—source data 1*.

The online version of this article includes the following source data and figure supplement(s) for figure 3:

**Source data 1.** In the absence of direct excitatory transmission from PFs, GABAergic-independent outward currents are preceded by a short-lasting inward current.

**Figure supplement 1.** Evidence for subthreshold excitatory postsynaptic current (EPSC) transmission across electrical synapses.

**Figure supplement 1—source data 1.** Evidence for subthreshold excitatory postsynaptic currents (EPSC) transmission across electrical synapses.

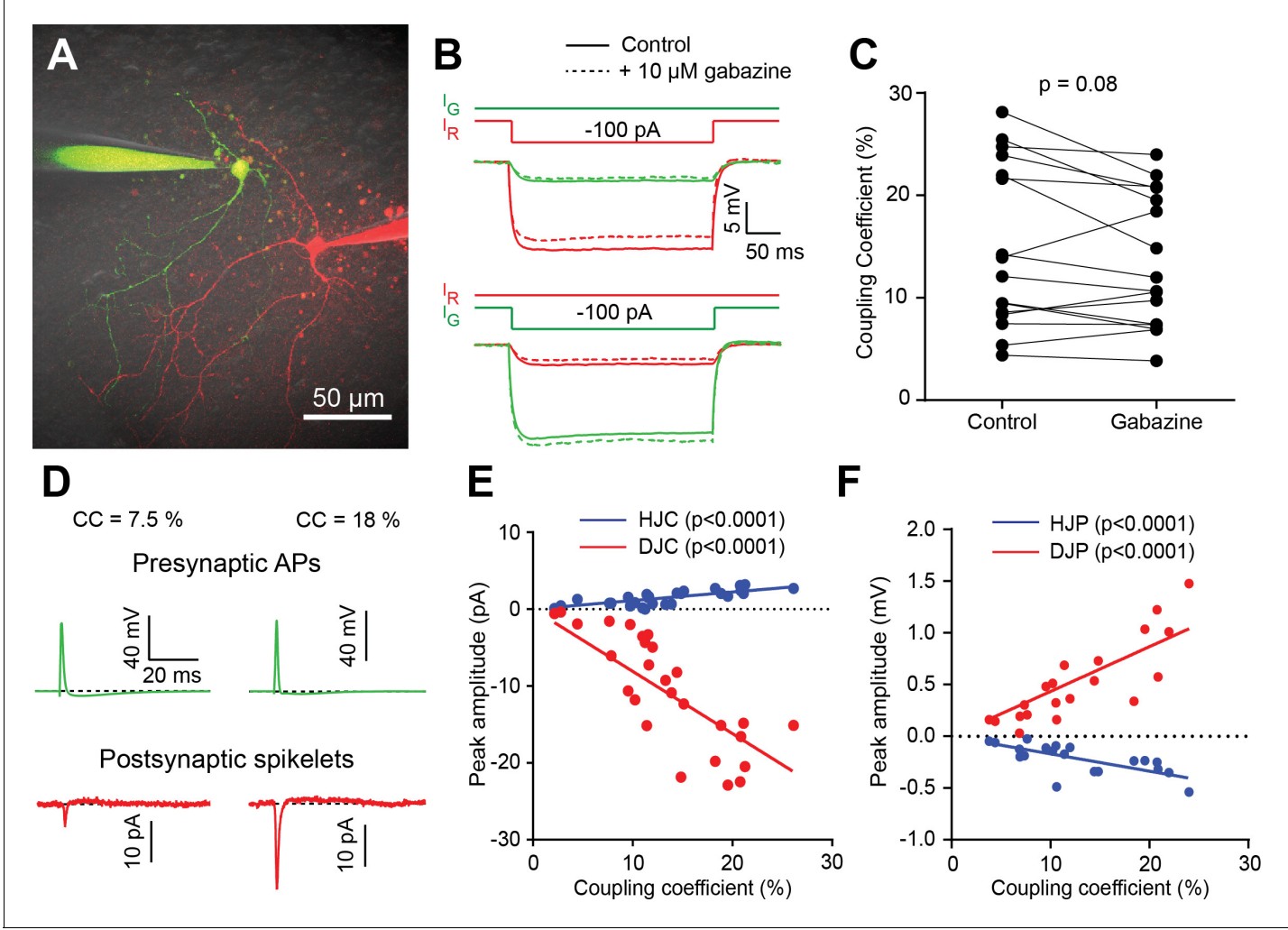

**Figure 4.** Electrical coupling strength between BCs correlates with spikelet amplitude. (**A**) 2PLSM image (maximal intensity projection) of two BCs loaded with 20 µM Alexa 594 (red), or 20 µm Alexa 488 (green) using dual whole-cell patch clamp. (**B**) Membrane potential of both cells (same color code as in A), when the red cell is injected with a hyperpolarizing current pulse (upper panel), or when the green cell is injected with the same current pulse (lower panel). Solid lines indicate responses in control conditions, while dashed lines correspond to responses recorded in 10 µM gabazine. Traces are averages of 50 sweeps. (**C**) Summary plot of unidirectional coupling coefficients (CCs), showing that they are not significantly altered by gabazine addition (n = 16 cells from 8 BC pairs, Wilcoxon matched-pairs signed rank test, p=0.08). (**D**) Two representative paired BC recordings, one with an average (left) and one with a high bidirectional CC (right), and their corresponding spikelet currents (red traces, holding membrane potential of −70 mV) resulting from an AP elicited in the presynaptic cells (green traces, RMP of −70 mV). (**E**) Summary plot of paired recordings showing the relationship between spikelet inward currents (depolarizing junction current, DJC, red), fAHP-mediated outward current (hyperpolarizing junction current, HJC, blue) and CC (n = 20 spikelet recordings). Note that in these recordings, presynaptic (transmitting) cells were maintained at approximately −70 mV, and 10 µM gabazine was present to block GABAergic inputs. Lines are linear regressions (n = 27 cells from 14 pairs; HJC: p<0.0001, DJC: p<0.0001) (**F**) Similar to B, but for spikelets recorded in current clamp (n = 27) DJP: depolarizing junction potential; HJP: hyperpolarizing junction potential. Lines are linear regressions (n = 20 cells from 10 pairs; HJP: p<0.0001, DJP: p<0.0001). Holding potentials were ~−70 mV. See also *Figure 4—source data 1*.

The online version of this article includes the following source data and figure supplement(s) for figure 4:

**Source data 1.** Electrical coupling strength between BCs correlates with spikelet amplitude.

**Figure supplement 1.** Relationship between coupling coefficients and spikelet transmission, and distance-independence of electrical coupling.

**Figure supplement 1—source data 1.** Relationship between coupling coefficients and spikelet transmission, and distance-independence of electrical coupling.

(*Figure 4D*). Indeed, the amplitudes of the inward and outward currents (or their corresponding synaptic potentials) correlated with the bidirectional CCs for each cell pair (*Figure 4D and E*). Given the number of trials (n = 30 sweeps), spikelets could generally be detected if the CCs were greater than 2% (see also *Figure 4—figure supplement 1A*). However, for small CCs between 2% and 5% we did not always observe a spikelet (*Figure 4—figure supplement 1B*), which could be explained by an electrical connection via an intermediate third MLI (*Kim et al., 2014*). These results argue strongly that spikelet transmission is mediated by the same gap junctions that generate electrical coupling.

## Modulation of spikelet polarity by presynaptic membrane potential

Spikelets recorded from different cell types throughout the brain have been shown to differ in their waveforms, and notably in the balance between excitation and inhibition (*Dugué et al., 2009*; *Galarreta and Hestrin, 2002*; *Hu and Agmon, 2015*). In immature BCs, a predominant depolarizing component was reported (*Alcami, 2018*; *Alcami and Marty, 2013*). Because of the prominent outward current, we more closely investigated the net polarity of spikelets and the influence of membrane potential using paired BC recordings. APs were triggered when varying the holding membrane potential of either the receiving BC (*Figure 5A*) or the transmitting BC (*Figure 5C*). The peak amplitude of the inward and outward currents, as well as the charge transfer of the spikelet response, were unaffected by altering the holding potential of the receiving neuron (*Figure 5B*, n = 12 cells from six pairs). In contrast, when adjusting the holding current to alter the membrane potential of the presynaptic neuron between approximately −80, −70 or −60 mV, the inward current component of the spikelet reduced in amplitude and the outward current component increased (*Figure 5D*; n = 12 cells from six pairs). Because of the slower outward component, the compound effect of inward and outward current changes led to an inversion of charge transfer from net excitation to net inhibition as the presynaptic neuron was depolarized. These results reveal the critical importance of presynaptic membrane potential on the net polarity of spikelets, and suggest that the ratio of inward/outward current peak amplitudes can be used to estimate the resting membrane potential of electrically coupled cells without the need for direct whole-cell recording.

In order to estimate the resting membrane potential of unperturbed electrical partners (*i.e.*, without whole-cell dialysis), we compared the peak inward and outward current amplitudes of spikelets from paired recordings when altering presynaptic membrane potential to those evoked by off-beam stimulation and direct stimulation (from *Figure 3*). A linear regression analysis allowed determination of the differences of inward-outward ratios for each condition (*Figure 5E*). In paired recordings, the slopes of the regression lines from the three presynaptic membrane potential groups (−60,−70 and −80 mV) were significantly different (F-test, p-value=0.0003), indicating that the ratio of inward and outward currents is a reliable indicator of presynaptic resting membrane potential. The regression line of the off-beam stimulation group was comparable to the −60 mV group from paired recordings (p-value=0.93), but the linear relationship was significantly different from the direct stimulation group and the −60 mV group (p-value=0.01). Provided that the voltage-dependence of the shape of the presynaptic AP is similar between patch-dialyzed and unperturbed cells, these data indicate that MLIs in brain slices have a depolarized resting membrane potential, greater than −60 mV (*i.e.* more depolarized). *Figure 5D* shows that spikelet responses from −60 mV holding potentials carry a net inhibition, as the total charge integral was positive. Because we were not certain to have only single neuron spikelet responses to off-beam and direct stimulation, we next normalized spikelets to the peak amplitude of the inward current before integration (over 50 ms) and then compared the net relative charge to that from spikelets from paired recordings when holding the presynaptic neuron at −60 mV (*Figure 5F*). Consistent with a net inhibitory effect, the relative time integrals for directly stimulated and off-beam recruited spikelets were positive. The net relative charge of off-beam-evoked spikelets was similar to paired recordings where the presynaptic neuron was held at −60 mV, but directly recruited spikelets displayed an even larger normalized inhibitory charge transfer. The difference between the off-beam-elicited and directly evoked spikelets could be due to the passage of subthreshold EPSCs transmitted from presynaptic neurons in the off-beam synaptic stimulation case, which contributes an additional inward current (See *Figure 3—figure supplement 1*) and an underestimation of the relative outward current.

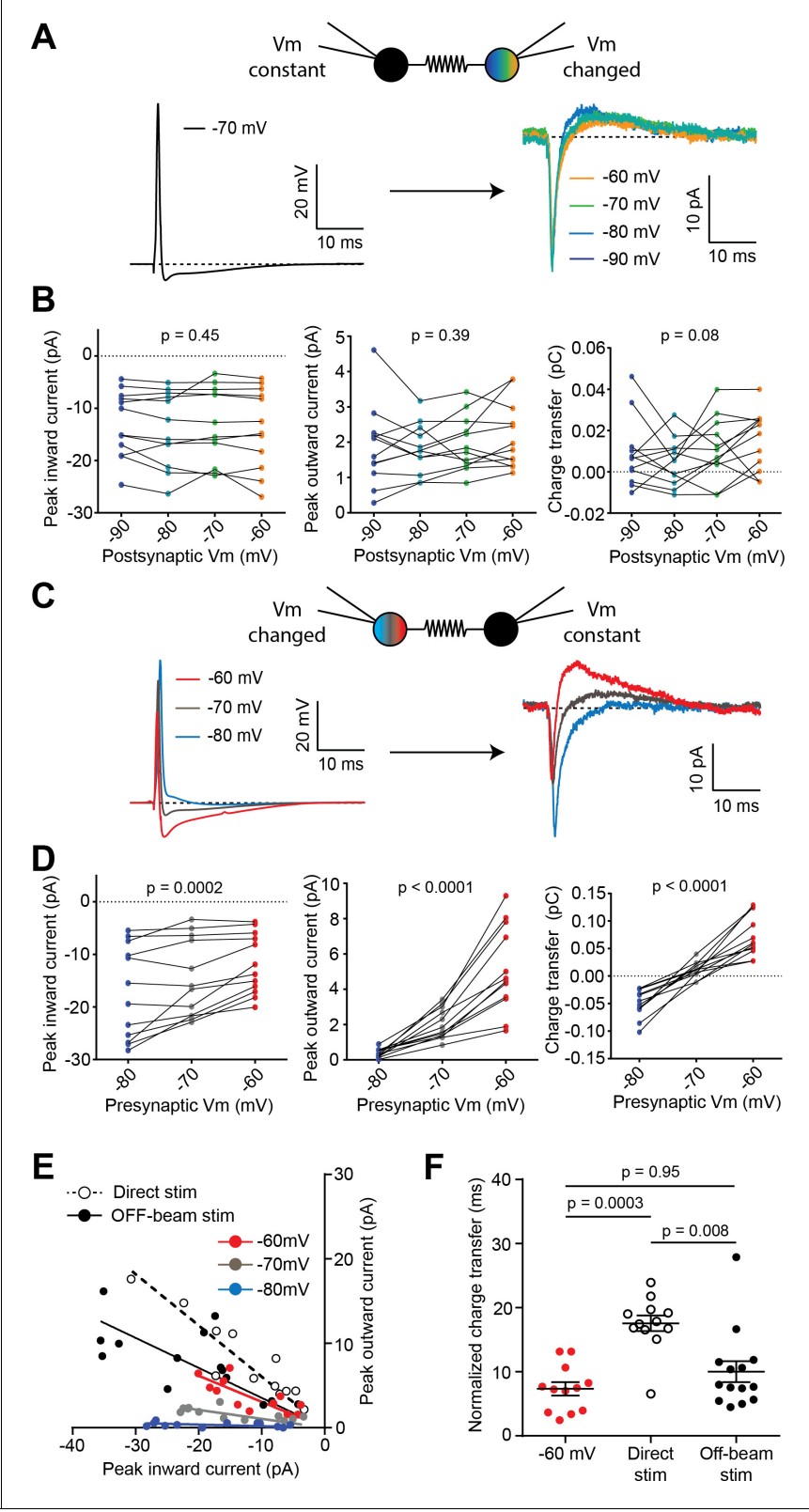

**Figure 5.** Presynaptic resting potential influences polarity of spikelets. (**A**) Action potential from one cell of a paired BC recording (left; holding potential = −70 mV) and corresponding spikelets current recorded in a voltage-clamped postsynaptic cell at different holding membrane potentials (Vm, right). (**B**) Inward current peak amplitude (left panel), outward current peak amplitude (middle panel) and charge transfer (right panel) are not significantly altered by changing postsynaptic Vm (n = 12 spikelets from 6 BC pairs, three independent Friedman tests, p-values from left to right:

*Figure 5 continued on next page*

Figure 5 continued

p=0.45, p=0.39, p=0.08). (**C**) Action potential waveforms from the same cell recorded at three different resting Vm (left), with corresponding spikelets recorded in the postsynaptic cell, held at −70 mV in voltage-clamp (right). (**D**) Inward current peak amplitude (left panel), outward current peak amplitude (middle panel) and charge transfer (right panel) are all significantly altered by changing the resting membrane potential of the presynaptic cell (n = 12 spikelets from six pairs, three independent Friedman tests). (**E**) Plot of outward versus inward current peak amplitudes of spikelets recorded under five conditions: either from pairs of connected BCs (Presynaptic Vm = −60,−70, −80 mV), from extracellular direct stimulation (dotted) or off-beam stimulation (black). Regression lines were performed on the individual groups. Slopes were significantly different between the −60, −70, and −80 mV groups (F-test, p-value=0.0003), significantly different between the direct stimulation and the −60 mV groups (p-value=0.013), but not different between the off-beam stimulation and −60 mV groups (p-value=0.93). (**F**) Summary plot of peak-normalized inward charge transfer of spikelets recorded in voltage clamp and elicited from a transmitting BC pair whose holding potential (Vm) was −60 mV, or from extracellular direct and off-beam stimulation (Kruskal-Wallis test with Dunn's multiple comparisons; −60 mV (Pair): n = 12 cells from six pairs, Direct stim.: n = 12 stimulation sites over 10 cells, off-beam stim.: n = 14 sites over 11 cells). See also *Figure 5—source data 1*.

The online version of this article includes the following source data for figure 5:

**Source data 1.** Presynaptic resting potential influences polarity of spikelets.

Altogether, these results indicate that the resting membrane potential of MLIs forming electrical synapses onto BCs is a critical determinant influencing the net excitatory/inhibitory impact of spikelet transmission. Moreover, we used the relative inward and outward ratio of spikelet currents to suggest that in mature mice, MLIs have a depolarized resting membrane potential (more than −60 mV), in agreement with previous estimates (*Chavas and Marty, 2003*; *Kim et al., 2014*). This depolarized membrane potential amplifies the AHP in the presynaptic neuron and ensures that spikelets deliver a net inhibition, which provides another mechanism of FFI within the MLI network, in addition to chemical transmission.

## Electrical synapses between BCs are more frequent than GABAergic synapses

In the inner-third MLI population of young rats, connections by electrical synapses were found to be more common than chemical ones (*Rieubland et al., 2014*). We re-examined the relative impact of electrical synapse-mediated FFI and classical chemical FFI in mature mice, as developmental changes have been described in other brain regions (*Hormuzdi et al., 2001*; *Peinado et al., 1993*). Previous studies used multi-electrode whole-cell recordings to establish the fraction of synapses that were chemical and electrical, albeit by using long current step injections and a criteria of >1% CC to identify electrically coupled cells (*Rieubland et al., 2014*). Here, electrical coupling was examined using both long hyperpolarizing current injections (see *Figure 4B*) and brief current injections to elicit APs in the presynaptic neuron while monitoring the postsynaptic voltage and current responses (*Figure 6B*). To probe chemical connectivity, we compared the postsynaptic responses to presynaptic AP generation, either before and after application of gabazine or at different holding membrane potentials of the postsynaptic cells (held at −70 mV or −60 mV, *Figure 6B*) in order to increase the driving force for GABAergic currents. As shown above, this manipulation does not influence the waveform of potential superimposed spikelets (*Figure 5A and B*). *Figure 6B* shows how each type of connection could be unambiguously identified: (1) unidirectional chemical connections did not display inward currents but displayed a voltage- and/or gabazine-sensitive outward current, (2) unidirectional spikelet transmission displayed inward currents along with voltage- and gabazine-insensitive outward currents, (3) dual connections displayed significant inward currents along with voltage- and gabazine-sensitive outward currents, and finally, (4) absence of synaptic connections was inferred when no postsynaptic response was observed (see a summary of conditions in *Figure 6—figure supplement 1*).

In order to compare the frequency of chemical and electrical synapses, we considered only electrical connections that showed spikelet transmission, since weak CCs may reflect indirect coupling between BCs (<2.3%; see *Figure 4—figure supplement 1A,B*). Since spikelet transmission across gap junctions is symmetrical (bidirectional), we counted maximally one electrical connection per paired BC recording. For a chemical connection, however, there could be zero, one, or two connections per paired recording, and thus we estimated the number of chemical synapses per connections tested. We found that bidirectional spikelet transmission was significantly more likely to be observed than unidirectional GABAergic connections, (58 ± 6%, n = 35/60 paired recordings, and 41 ± 5%, n =

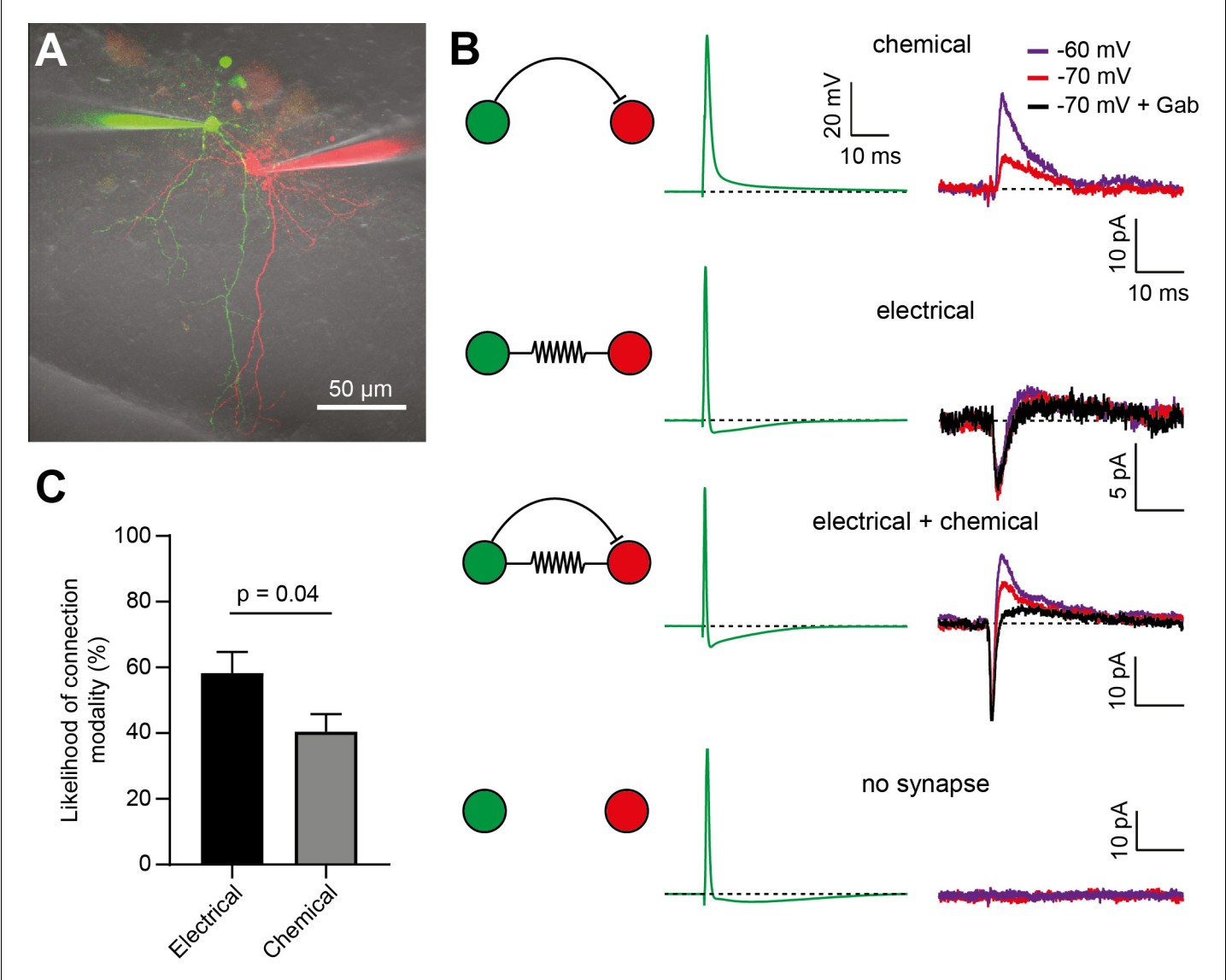

**Figure 6.** Electrical synapses are more frequent than chemical inhibitory synapses between BCs. (**A**) 2P-LSM image (maximal intensity projection) of two BCs loaded with 20 μM Alexa 594 (red) or 20 μm Alexa 488 (green). (**B**) Representative examples of each of the 3 types of unidirectional connections. Presynaptic AP waveforms are shown in green, with the corresponding responses observed in the postsynaptic cells shown on their right (black, purple or red). Chemical synapses were identified by the presence of a postsynaptic Vm-sensitive or gabazine-sensitive outward current (Gab). Electrical synapses were identified by the presence of a spikelet-mediated inward current. The fourth condition includes those paired recordings with no evidence of electrical or chemical synapses (no synapse). (**C**) Bar graph showing that electrical synapses are significantly more frequent than chemical synapses in the BC network. 35 out of 60 pairs showed bidirectional spikelets (electrical synapses), whereas 35 out of 85 unidirectional connections showed evidence of chemical synapses (unpaired t-test assuming binomial distributions, p=0.04). See also *Figure 6—source data 1*. The online version of this article includes the following source data and figure supplement(s) for figure 6:

**Source data 1.** Electrical synapses are more frequent than chemical inhibitory synapses between BCs.

**Figure supplement 1.** Identification of GABAergic connections in BC paired recordings.

**Figure supplement 1—source data 1.** Identification of GABAergic connections in BC paired recordings.

35/85 connections, respectively; p=0.04, unpaired t-test). If we analyzed the gabazine experiments only, we found that in 15 of 28 unidirectional connections from BC-BC pairs, the GABA$_A$R antagonist did not alter the outward current amplitude, consistent with a larger fraction of electrical than chemical connections. In the remaining connections the fractional block by 10 μM gabazine was ~50% on average (*Figure 6—figure supplement 1B*), suggesting that, when both chemical and electrical

synapses were present, the outward current under our recording conditions was similar. We cannot rule out the possibility that GABAergic transmission is more sensitive to rundown and might bias the estimate of the relative contribution of electrical and chemical synapses. Nevertheless, our data suggest that, in adult animals, electrical synapses between BCs are more likely to mediate FFI than their chemical counterparts.

## FF recruitment of spikelets narrows single EPSP width and dampens temporal summation

Having observed that electrical synapses are a prominent source of inhibition between BCs, we investigated whether the FF recruitment of spikelets could influence EPSP kinetics and their temporal summation, as for chemical FFI. Because mefloquine is known to alter input resistance by blocking electrical synapses (*Vervaeke et al., 2012*) and has been shown to alter membrane capacitance (*Szoboszlay et al., 2016*), we designed a specific experiment to examine the influence of spikelets on EPSPs without the need for gap junction antagonists. Taking advantage of the fact that some stimulation locations along the dendrite did not recruit a spikelet when stimulating 5–10 V above threshold to elicit an EPSC (low stimulation, LS; see *Figure 1*), we examined EPSP shape under low and high extracellular stimulation intensities (+15 V; HS), which in some cases recruited AP firing of a neighboring interneuron and thus a spikelet in the recorded BC. Recordings were pooled into two different groups: (1) one in which no electrical neighbor was recruited in either stimulation regime (*Figure 7A*), and (2) another in which increasing stimulation intensity was sufficient to recruit an electrical neighbor (*Figure 7C*). For those cases where an outward current was *not* recruited, current-clamp recordings of the same cells showed no differences in the half-width of the EPSPs (n = 15, p=0.19; *Figure 7B*), despite an increase in the peak EPSC amplitude. For the cases where additional stimulation intensity elicited a detectable outward current, the half-width in current-clamp conditions was decreased by 21 ± 4% (n = 9, p=0.004; *Figure 7D*). The variability in the EPSC decay across stimulation locations in response to low-intensity stimulation could be due to the variability in cable filtering when eliciting synapse stimulation in different locations within the dendrite, or the variability in the number of electrically coupled cells, which have been shown to decrease input resistance (*Alcami and Marty, 2013*). But we cannot rule out that a small spikelet may accelerate some EPSPs without eliciting a detectable outward current. Nevertheless, recruiting an additional spikelet, on average, accelerated EPSP decays. These results support the notion that FF recruitment of spikelets sharpens the temporal precision of EPSPs, as is the case for classical, chemical FFI.

To examine the influence of FF spikelet recruitment on the temporal summation of EPSPs, we applied high-frequency train stimuli at dendritic locations with or without detectable outward currents. EPSPs with spikelets produced faster compound EPSPs followed by a hyperpolarizing component, as compared to those without spikelets (*Figure 7E*). Temporal summation of EPSPs in response to five stimuli at 50 Hz was significantly reduced at stimulus locations that recruited spikelets (*Figure 7F*; n = 27 stimulation sites in the 'no spikelet' group versus n = 22 stimulation sites in the 'spikelet' group; p=0.004, Two-way repeated measures ANOVA). This difference could neither be attributed to differences in peak amplitude of the first EPSP, nor to uneven sampling of PF-mediated responses along the dendritic trees (*Figure 7—figure supplement 1*). Together, these data indicate that spikelets can act like chemical FFI by reducing the half-width and dampening the temporal summation of EPSPs.

## Spikelet signaling enables temporal contrast enhancement of transient excitation

Because spikelets are also comprised of a brief depolarizing component, we considered the possibility that they could contribute to enhanced firing rates over a brief window, in contrast to classical chemical FFI that reduces firing rates. The challenge was to generate EPSPs and spikelets independently, and without altering the EPSP amplitude (as in *Figure 7*), in order to examine the influence of spikelets on synaptic integration and EPSP-spike coupling. We, therefore, recorded EPSPs in single BCs following stimulation of two independent PF pathways: one electrode was positioned above the BC dendritic tree (on-beam) to recruit only direct EPSPs; and another was positioned adjacent to the projected plane of the dendritic tree to stimulate off-beam, so as to only recruit spikelets in the recorded cell (as in *Figure 3*; *Figure 8A–C*). Stimulation intensity was adjusted independently for

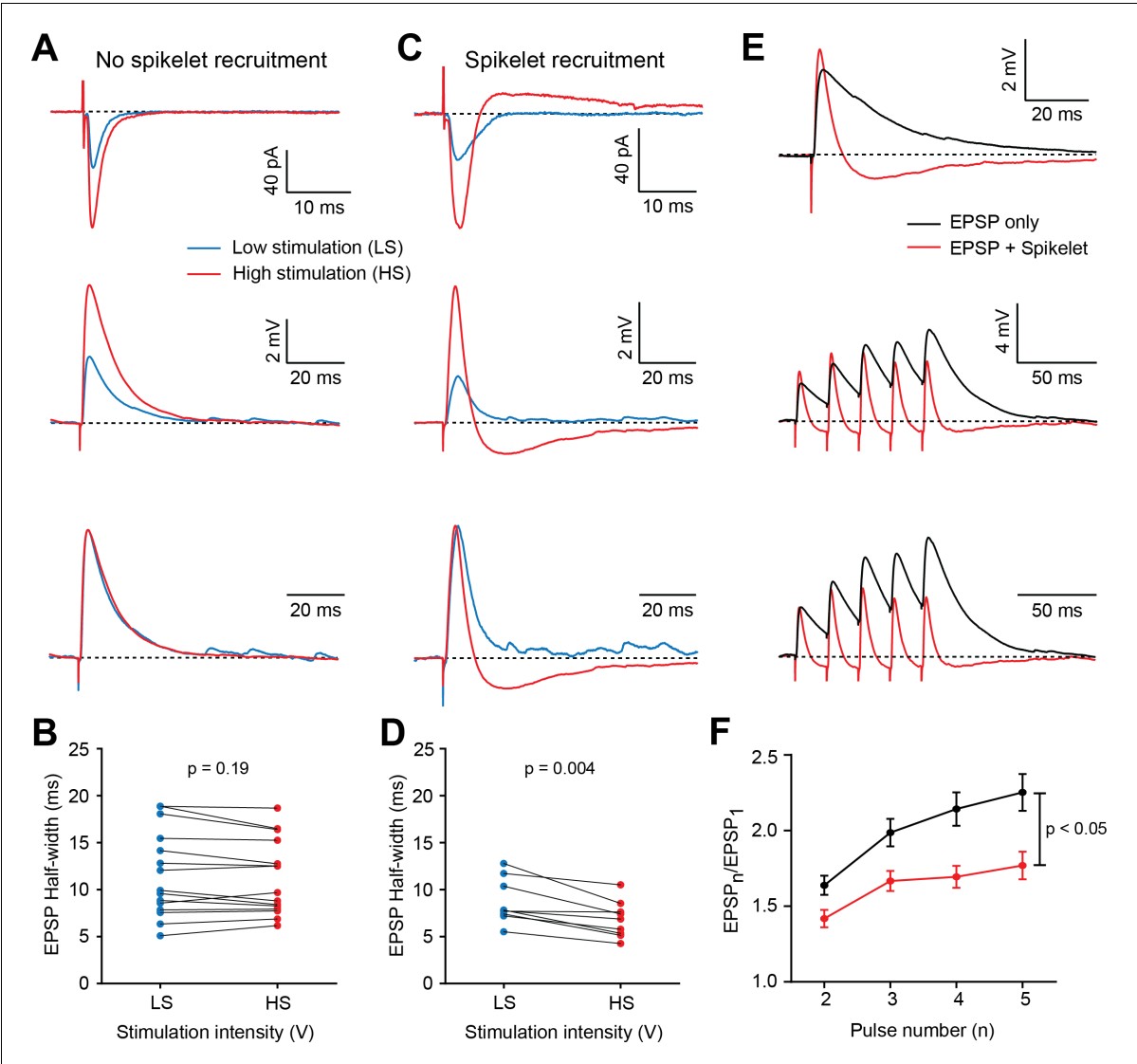

**Figure 7.** Feed-forward (FF) recruitment of spikelets narrows excitatory postsynaptic potential (EPSP) width and dampens temporal summation. (**A**) Top, voltage-clamp recordings of PF-mediated synaptic responses in a representative BC in response to low stimulation intensity (LS – blue traces) and high stimulation intensity (HS – red traces). Increasing stimulation intensity leads to increased peak amplitude of excitatory postsynaptic currents (EPSCs), with no additional recruitment of spikelets. Middle, current-clamp recording of the same cell, showing that increasing stimulation intensity increased the amplitude but not the half-width of the normalized EPSPs (bottom). Traces are averages of 20 to 30 trials. (**B**) Summary plot showing no increase in half-width of EPSPs if no outward current was observed in voltage clamp (p=0.19, Wilcoxon matched-pairs signed rank test, n = 15 stimulation sites over 12 cells). (**C**) Example cell in which increasing the stimulation intensity recruits an additional spikelet, observed in voltage clamp (top) and current clamp (middle). The half-width of normalized EPSPs was shortened. (**D**) Summary plot showing a decrease in half-width of EPSPs if outward currents were observed in voltage clamp (p=0.004, Wilcoxon matched-pairs signed rank test, n = 9 sites from 8 cells). (**E**) Top, superimposed averaged postsynaptic responses (n = 30 trials) from two different cells: one recorded in response to LS, and another in which the direct EPSP is followed by a spikelet-induced hyperpolarization (red) and the other is not (black). Middle, EPSPs evoked by a 50 Hz stimulus train in the same cells as above. Note the decreased temporal summation in the group where spikelets are present. (**F**) Summary plot showing relative EPSP amplitudes in response to all five stimuli for stimulation sites that did (red) or did not (black) evoke spikelets (Two-way repeated measures ANOVA, column factor (with versus without spikelet): p=0.004; interaction factor (difference in curve shape with stimulus number): p=0.001; EPSP only: n = 27 stimulation sites over 16 cells, EPSP with spikelet: n = 22 sites over 17 cells). See also *Figure 7—source data 1*.

The online version of this article includes the following source data and figure supplement(s) for figure 7:

**Source data 1.** Feed-forward(FF)recruitment of spikelets narrowsexcitatory postsynaptic potential (EPSP)width and dampens temporal summation.
**Figure supplement 1.** Control experiments for spikelet effect on excitatory postsynaptic potentials (EPSPs).
**Figure supplement 1—source data 1.** Control experiments for spikelet effect onexcitatory postsynaptic potentials(EPSPs).

each location to achieve on- and off-beam stimulation. Stimulation pipette positions were always displaced by at least 100 μm, to ensure the recruitment of two different PF beams. BCs were initially held between −75 and −70 mV in order to examine the amplitude of the compound subthreshold synaptic response with respect to the time delay between the two stimuli (*Figure 8D*). Coincident stimulation of spikelets and EPSPs indeed increased the EPSP peak amplitude (averaged over a 4–6 ms window) by 21.3 ± 2.4% (n = 12, p<0.0001, Wilcoxon matched-pairs signed rank test - *Figure 8E*). However, if spikelet recruitment preceded the EPSP (by 5 to 75 ms), the peak amplitude of the EPSP was significantly reduced, with a maximal reduction of 20 ± 2.9% at Δt = 20 ms (n = 12, p<0.0001, Wilcoxon matched-pairs; *Figure 8E*). We also confirmed that spikelets concomitant with EPSPs reduced their half-width by 16.7 ± 1.8%, consistent with results described above in *Figure 7D*.

Finally, we examined if these changes in peak amplitude of compound synaptic responses could translate into differences in spike probability. We injected current to maintain the holding membrane potential at −65 and −60 mV, resulting in EPSP-induced AP firing in approximately 50% of the trials (*Figure 8F*). We then examined AP firing probability versus time delay between on- and off-beam stimulation. We found that the relative spike probability was increased by 49.3 ± 8% for coincidental arrival of both inputs, and decreased by 40.5 ± 6.1% if EPSPs were triggered 20 ms after the spikelets (p-values<0.001; n = 11 cells for both; Wilcoxon matched-pairs; *Figure 8G*). Thus, a unique property of spikelet-mediated FF modulation is the ability to enhance spike probability during brief time windows and decrease it for longer ones, a form of temporal contrast enhancement of information conveyed by spatially clustered axonal beams (*Wilms and Häusser, 2015*). Moreover, for independent PF beams, the MLI network can detect coincident excitation that will in turn generate a global and precise inhibition of downstream PCs.

## Discussion

MLIs within the cerebellar cortex have been shown to provide FFI onto PCs in vitro (*Dizon and Khodakhah, 2011*; *Mittmann et al., 2005*; *Valera et al., 2016*) and in vivo (*Arlt and Häusser, 2020*; *Blot et al., 2016*). MLIs of the cerebellar cortex are known to communicate through both chemical and electrical synapses (*Kondo and Marty, 1998*; *Mann-Metzer and Yarom, 1999*; *Rieubland et al., 2014*). GABAergic chemical synapses provide self-inhibition between MLIs (*Mittmann et al., 2005*), and have been suggested to mediate disinhibition of PCs (*Blot et al., 2016*). Whether or not electrical connectivity could be recruited in a similar manner and influence EPSP-spike coupling, however, had not yet been examined. Here, we identify and describe the functional implication of FF recruitment of spikelet signaling in cerebellar BCs, an MLI subtype that exhibits a high probability of electrical synaptic connections (*Alcami and Marty, 2013*; *Rieubland et al., 2014*). Our results indicate that extracellular electrical stimulation of small PF beams leads to the recruitment of APs in electrically coupled MLIs that generate spikelets in the postsynaptic BCs (*Figures 1–3*), but more rarely and less prominently in SCs (*Figure 2*). Because of the low frequency (13%) of BC-SC electrical coupling (*Rieubland et al., 2014*), we propose that eFFI is mostly provided by other BCs. We show that the polarity of spikelet responses is net inhibitory and can be influenced by the presynaptic neuron's resting membrane potential (*Figure 5*). This type of connection is more frequent than those of chemical inhibition (58% vs. 41%; *Figure 6*). Moreover, the brief depolarizing component of the spikelet provides a distinct advantage over cFFI because the postsynaptic output firing can be briefly enhanced, providing a temporal contrast enhancement (*Figure 8*). Such a mechanism generates strong synchronized inhibition from BCs that could temporally entrain PCs that in turn drive deep cerebellar nuclei (*Brown and Raman, 2018*; *Özcan et al., 2020*; *Person and Raman, 2012*).

### Cellular mechanism of spikelet transmission

Extracellular stimulation of PFs in parasagittal slices in the presence of gabazine produced outward currents on average of 10 pA in BCs and less than 3 pA in SCs (*Figures 1* and *2*). We performed several experiments whose results were consistent with spikelet transmission through gap junctions as the origin of GABAergic-independent outward currents: (1) subsaturating concentrations (20 μM) of the gap junction blocker mefloquine blocked the outward current by 70%, as expected for a current entirely mediated by gap junctions (*Cruikshank et al., 2004*). (2) We observed a 3-fold reduction in

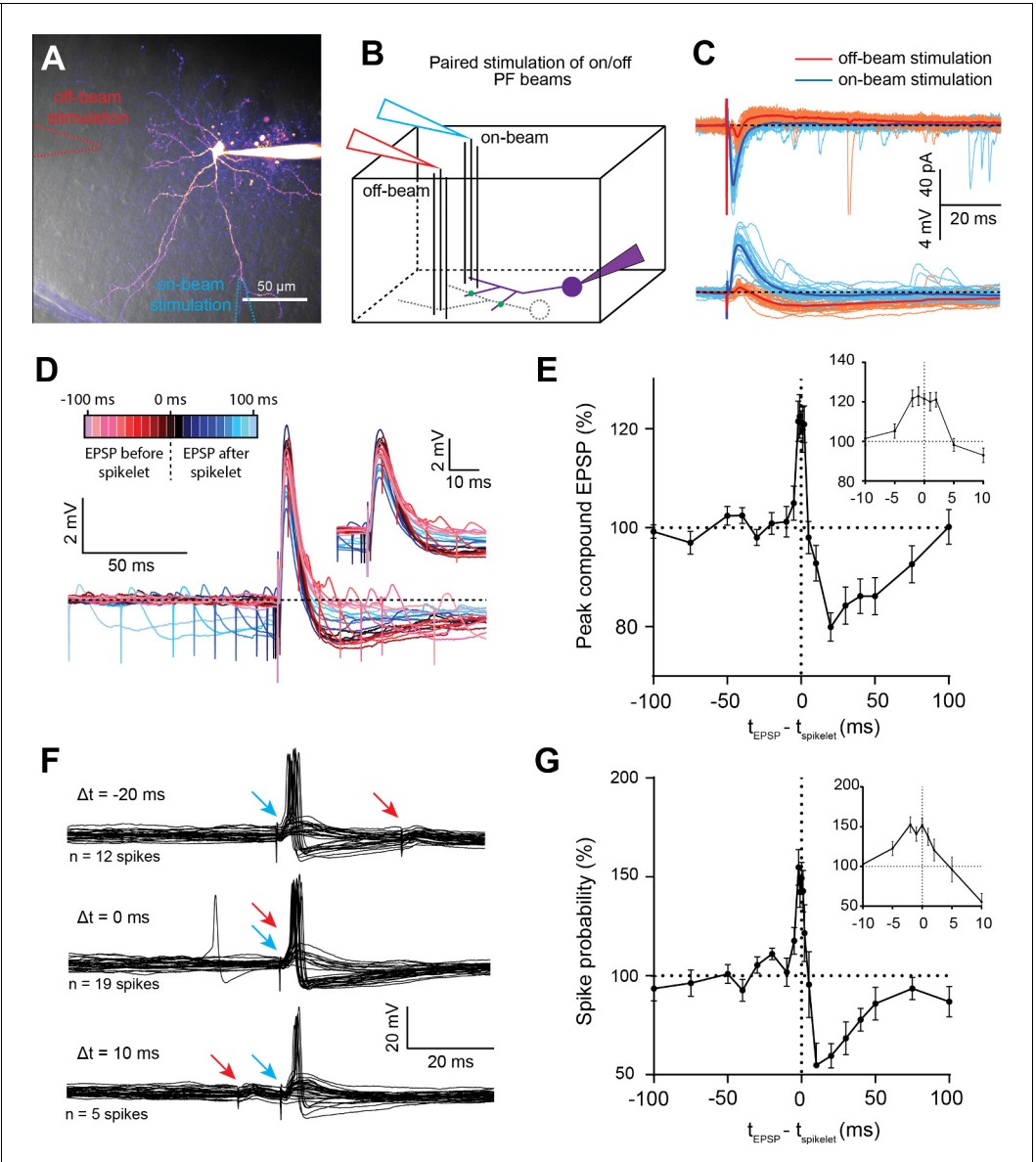

**Figure 8.** Spikelet transmission enables temporal contrast enhancement of basket cell (BC) excitation. (A) 2PLSM image (maximal intensity projection) of a BC loaded with 20 µM Alexa 594. Dashed lines indicate the position of the stimulating pipettes on top of the slice (off-beam stimulation in red, on-beam stimulation in blue). (B) Parasagittal block diagram showing on- (blue) and off-beam (red) extracellular electrode stimulation. (C) Superimposed voltage-clamp recordings (upper panel) and corresponding current-clamp recordings (lower panel) of postsynaptic responses to on-beam stimulation (light blue), or off-beam stimulation to elicit a spikelet (orange). Dark blue and red traces represent the averages of 30 trials for on and off-beam stimulation, respectively. (D) Superimposed compound excitatory postsynaptic potentials (EPSPs) recorded in current clamp and for different time delays between on- and off- beam stimulation (each trace is an average of 25 to 30 trials). Time delays (Δt) are relative to on-beam stimulation times. Positive time delays (*i.e.*, off-beam stimulation before on-beam stimulation) are shown in blue shades, coincident stimulations are shown in black, and negative time delays are shown in red shades. Inset is an expanded time scale. (E) Peak amplitude of the compound synaptic responses (normalized to the mean amplitude of the single EPSPs alone) versus the time delay between EPSP and spikelet recruitment (n = 12 cells). Inset is an expanded timescale around the peak and in units of milliseconds. (F) Single trial AP responses to different delays between on- and off-beam stimulation. Shown here are three representative cases: spikelet (red arrows) recruited 20 ms after EPSPs (blue arrows) (top panel), coincidentally with EPSPs (middle panel), or 10 ms before the EPSP (lower panel). In each instance, 30 sweeps are shown. (G) Summary plot of the probability of eliciting an AP by on-beam stimulation (normalized to the mean probability of AP firing by on-beam stimulation alone) versus the time delay between EPSP (on-beam stimulation) and spikelet recruitment (off-beam stimulation; n = 11 cells). Inset is an expanded timescale around the peak and in units of milliseconds. See also *Figure 8—source data 1*.

The online version of this article includes the following source data for figure 8:

**Source data 1.** Spikelet signaling transmission enables temporal contrast enhancement of basket cell (BC) excitation.

peak amplitude of the outward current in Cx36 KO mice (*Figure 2*). The incomplete knockdown could be due to genetic compensation by Cx45, which is known to also be expressed in MLIs (*Van Der Giessen et al., 2006*). Nevertheless, we only observed 20% of BC-BC pairs in Cx36 KO mice that showed bidirectional spikelets and a CC greater than 2% (the average CC correlated with the observation of a spikelet, *Figure 4—figure supplement 1A*), as opposed to the nearly 60% of BC-BC pairs in WT (*Figure 6C*). (3) To confirm that the outward currents were mediated by spikelets, we showed that stimulation of PFs outside the dendritic field of the recorded cell and direct stimulation of MLIs in NBQX both evoked currents with a rapid inward component and an outward component similar to that following EPSCs. (4) Moreover, these extracellular stimulation-evoked spikelets were very similar to those evoked by single APs propagating between electrically coupled BC pairs. And finally, (5) the strength of electrical coupling between two BCs, estimated from 400 ms step depolarizations, was positively correlated with spikelet amplitude (*Figure 4*), suggesting that spikelets were mediated by gap junctions.

Electrical coupling between neurons has also been demonstrated in the absence of gap junctions or chemical synapses and is referred to as ephaptic coupling. We cannot directly rule out that the remaining spikelet currents in mefloquine or Cx36 KOs could result from transmission of an AP between closely juxtaposed dendrites or axons. BC-PC ephaptic transmission is thought to occur between the axon terminal of BCs (pinceau) and the axon initial segment of PCs. Both the capacitive and K-currents generate a local field that sufficiently hyperpolarizes the AIS membrane potential and reduces the probability of PC spiking (*Blot and Barbour, 2014*). In that study, simulations suggest that the ephaptic-mediated hyperpolarization is ~1 μV at the somata, and thus unlikely to account for the large mV changes we observe between two BCs. In contrast, ephaptic currents in PC-PC pairs are 10's of pA, due to a regenerative activation of local AIS sodium conductances in postjunctional PCs that is eliminated by postjunctional PC hyperpolarization to −70 mV (*Han et al., 2018*). We showed that the synapse-evoked outward current was insensitive to 1 mM of the sodium channel blocker QX-314 (*Figure 1*, see Materials and methods); and, in BC-BC paired recordings, the amplitude of the spikelet is insensitive to postsynaptic manipulation of membrane potential (*Figure 5*). These findings are inconsistent with a sodium-dependent ephaptic mechanism.

## Electrical synapse-mediated spikelets dynamically modulate BC EPSP-spike coupling

Spikelet waveforms have been shown to depend on a variety of physiological parameters across different cell types, including resting membrane potential (*Mann-Metzer and Yarom, 1999*; *Otsuka and Kawaguchi, 2013*) and active conductances in the presynaptic (*Curti and Pereda, 2004*; *Pereda et al., 2013*; *Russo et al., 2013*) or the postsynaptic cell (*Dugué et al., 2009*; *Mann-Metzer and Yarom, 1999*). By systematically varying the pre- and postsynaptic holding membrane potentials, we found that the latter had no detectable influence on the spikelet waveform (*Figure 5*). However, the presynaptic membrane potential is critical in shaping the triggered AP, and therefore the shape of the corresponding spikelet. A hyperpolarized resting potential leads to a net excitatory spikelet, while a depolarized resting potential reveals a prominent AHP, causing the corresponding spikelets to transmit net inhibition. Comparison of spikelets evoked in an unperturbed, electrically coupled neighbor to those evoked in paired recordings with presynaptic membrane potential manipulations is consistent with a depolarized BC resting membrane potential (> −60 mV; assuming voltage-dependence of AP shape is equal between unperturbed and dialyzed recordings). We postulate that previous recordings showing mostly depolarizing spikelets in cerebellar MLIs were due to either hyperpolarized holding potentials (*Alcami, 2018*) or an intrinsic difference in the amplitude of the AHP recorded from brain slices prepared from P11-13 rats.

Consistent with a net inhibitory action, we showed that FF recruitment of spikelets narrows the half-width of concomitant EPSPs and dampens their temporal summation. However, because the spikelets comprised a significant depolarizing component, perhaps due to the strong average CC (9.5 ± 1%; n = 41 pairs with CC >2.3%), we also examined their ability to enhance EPSP depolarization. When stimulating PFs within the dendritic tree of BCs, inward current components of spikelets were not reliably detectable. Nevertheless, we found that the transient excitatory drive of spikelets significantly increased the peak amplitude of the compound synaptic potential when both signals were coincident (within a 4–6 ms window). Thus, unlike cFFI, electrical feed-forward modulation (eFFM) shortens the temporal window for EPSP-spike coupling, thus

improving spike precision, while simultaneously enhancing spike probability (*Figure 8*). The degree of such temporal contrast enhancement could be enhanced by cFFI or regulated by factors that adjust presynaptic spike shape, in particular the relative size of the AHP, and thus regulate the temporal balance of excitation and inhibition. We observed that despite the net inhibitory drive of BC spikelets, they also show a prominent brief inward (depolarizing) current component. This depolarizing component of the spikelet effectively provides FF excitation and is thus capable of synchronizing electrically coupled interneurons (*Alcami, 2018*; *Mann-Metzer and Yarom, 1999*), whereas the hyperpolarization component could generate low-frequency resonance in a resting state of the network (*Dugué et al., 2009*) and/or desynchronize network activity upon synaptic stimulation (*Vervaeke et al., 2010*).

Here we did not examine the combined contribution of cFFI and eFFI, as its true contribution requires an accurate estimate of the GABAR reversal potential, which has not been performed in mature MLIs. Moreover, differential recruitment of cFFI and eFFI in MLIs under physiological spatiotemporal PF activation will require future experiments in awake animals. Nevertheless, we found that BCs are more likely to transmit spikelets than chemical inhibition to their neighbors (58 versus 41%, *Figure 6*), which could account for the narrowed excitatory responses of BCs versus SCs in vivo (*Chu et al., 2012*). Recent in vivo findings show that BC spikes have a stronger influence on PC spiking than SCs (*Arlt and Häusser, 2020*), implying that the specific timing of BCs will strongly influence PC output.

## Electrical connectivity in the cerebellar cortex is tuned for precise and synchronous output firing

We observed that 60% of PF stimulus pipette locations within the BC dendritic tree at P30-60 (nearly 80% of locations in P90-150 animals – *Figure 2*) produced sufficient excitatory drive in electrically coupled BC neighbors to reach threshold for APs, and thus generate a spikelet in the postsynaptic cell. This robust recruitment can be accounted for by three factors: (1) overlap of MLI dendritic trees in order to more likely receive common synaptic input (see *Figure 6A*), (2) a resting membrane potential close to threshold (*Figure 5E*; *Kim et al., 2014*), and (3) a strong enough CC, sufficient to transmit detectable spikelets (>2.3%; *Figure 5* and *Figure 4—figure supplement 1*). The mean CC was 9.5%, similar to the 12% found in Golgi cells (*Szoboszlay et al., 2016*).

This high degree of electrical connectivity can provide a global network mechanism spanning >100 µm in the sagittal plane (*Rieubland et al., 2014*) for generating a FF modulation of PC spiking patterns through synchronous inhibition (*i.e.*, excitation of MLIs) followed by a rapid disinhibition (*i. e.*, MLI self-inhibition). In support of the sagittal propagation of spikelets, we find no correlation in CC over intersomatic distances up to 60 µm (*Figure 4—figure supplement 1C*). Moreover, in vitro experiments revealed that electrical connectivity in the MLI population increases convergence from MLIs to PCs (*Kim et al., 2014*). Finally, the lateral spread of PF EPSPs through electrical synapses (*Figure 3—figure supplement 1*), as previously observed in Golgi cells (*Vervaeke et al., 2012*), along with spikelet transmission, would also contribute to increasing the effective convergence between MLI and PC.

In one of the few studies examining the activity of BCs and PCs in vivo (*Blot et al., 2016*), a biphasic correlogram suggested an initial excitation of PCs followed by a brief (8 ms) inhibition of PC firing. Recent double-patch recordings in vivo show that even a single deep MLI (likely BC) can drive detectable inhibition of PC firing (*Arlt and Häusser, 2020*). This very narrow and powerful inhibition could indeed be the result of precisely synchronized BC activity followed by self-inhibition, both mediated by eFFM. Such a network-wide synchronization would be less efficient via chemical inhibition, which has no depolarizing component and would be more variable from trial to trial due to the quantal properties of chemical transmission. Interestingly, voltage-clamp recordings in vivo showed that BC EPSCs are much faster than either SC or PC EPSCs in response to air puffs to the whisker pad (*Chu et al., 2012*), consistent with the presence of electrical synapses, and perhaps explaining the predominant inhibition of PC firing in response to sensory stimuli (*Bosman et al., 2010*; *Chen et al., 2016*). Thus, eFFM could ensure an enhanced and brief BC network recruitment necessary for precise refinement of PC firing patterns.

## Implications of eFFM in fine-tuning cerebellar-dependent motor behaviors

Whisker movement kinematics can be both encoded (*Chen et al., 2016*) and driven by precise PC firing patterns (*Bosman et al., 2010*; *Heiney et al., 2014*). This has been hypothesized to be due to millisecond synchrony of PC firing, which could be sculpted by inhibition from synchronously active BCs. Indeed, coherent MLI spiking is necessary to generate well-timed licking behaviors in mice (*Gaffield and Christie, 2017*). Synchronized PC activity can then precisely gate deep cerebellar nuclear neurons (*Özcan et al., 2020*; *Person and Raman, 2012*) to drive movement (*Brown and Raman, 2018*). Electrical synapses between BCs could provide a network-amplified, highly synchronous inhibition onto PCs along the parasagittal plane, despite complex spatio-temporal activation patterns of PFs. In turn, millisecond inhibition of PC firing by sensory stimuli can disinhibit cerebellar nuclei, which in turn drive motor responses (*Brown and Raman, 2018*). Finally, in vivo evidence that BCs are involved in refining PC firing on the millisecond scale supports their important role in timing cerebellar cortical output (*Arlt and Häusser, 2020*; *Blot et al., 2016*). We therefore propose that eFFM could be an important mechanism to achieve robust, globally synchronous and precisely timed control of PC firing that in turn can efficiently drive cerebellar output from the deep nuclei.

# Materials and methods

### Key resources table

| Reagent type (species) or resource | Designation | Source or reference | Identifiers | Additional information |
|---|---|---|---|---|
| Strain, strain background (*Mus musculus* – male and female) | CB6F1 | Mouse Genome Informatics | RRID:MGI:5649749 | |
| Strain, strain background (*Mus musculus* – male and female) | CX36 KO, 129 brown mice (129S1/SvlmJ) | David Paul, Harvard University | Deans et al. Neuron 2001. | The Cx36 coding sequence is replaced by a LacZ-IRES-PLAP reporter cassette |
| Chemical compound, drug | D-AP5 | Abcam | Cat#:ab120003 | |
| Chemical compound, drug | Gabazine (SR-95531) | Abcam | Cat#:ab120042 | |
| Chemical compound, drug | QX-314 | Abcam | Cat#:ab120118 | |
| Chemical compound, drug | NBQX | Tocris | Cat#:1044 | |
| Chemical compound, drug | Alexa Fluor 488 | ThermoFisher Scientific | Cat#:A10436 | |
| Chemical compound, drug | Alexa Fluor 594 | ThermoFisher Scientific | Cat#:A10438 | |
| Chemical compound, drug | Mefloquine hydrochloride | Sigma-Aldrich | Cat#:M2319 | |
| Chemical compound, drug | DMSO | Sigma-Aldrich | Cat#:D2650 | |
| Software, algorithm | Igor Pro | Wavemetrics | RRID:SCR_000325 | |
| Software, algorithm | Neuromatic | *Rothman and Silver, 2018*; DOI: 10.3389 | RRID:SCR_004186 | |
| Software, algorithm | ImageJ | National Institutes of Health | RRID:SCR_003070 | |
| Software, algorithm | GraphPad Prism 6 | GraphPad Software | RRID:SCR_002798 | |

### Slice preparation

Animal experiments were performed in accordance with the guidelines of Institut Pasteur, France, and all protocols were approved by the Ethics Committee #89 of Institut Pasteur (CETEA; approval # DHA180006). Experiments were performed on acute brain slices prepared from postnatal day 30–60 (~44 days on average) WT CB6F1 mice, which were bred in house from a cross between female BALB/cByJ and male C57BL/6J obtained from Charles River Laboratories. Experiments in *Figure 2* were performed on P90-150 Cx36 KO mice (129S1/SvlmJ genetic background in which the Cx36 coding sequence was replaced by a LacZ-IRES-PLAP reporter cassette), generated by Dr. David L.

Paul (Harvard University), and age-matched WT CB6F1 mice. All experiments were performed on both female and male mice without bias.

Acute parasagittal slices (200 μm) of cerebellar vermis were prepared as follows: Mice were rapidly killed by decapitation, after which the brains were removed and placed in an ice-cold solution containing (in mM): 2.5 KCl, 0.5 $CaCl_2$, 4 $MgCl_2$, 1.25 $NaH_2PO_4$, 24 $NaHCO_3$, 25 glucose, 230 sucrose, and 0.5 ascorbic acid. The solution was bubbled with 95% $O_2$ and 5% $CO_2$. Slices were cut from the dissected cerebellar vermis using a vibratome (Leica VT1200S), and incubated at 32°C for 30 min in a solution containing (in mM): 85 NaCl, 2.5 KCl, 0.5 $CaCl_2$, 4 $MgCl_2$, 1.25 $NaH_2PO_4$, 24 $NaHCO_3$, 25 glucose, 75 sucrose and 0.5 ascorbic acid. Slices were then transferred to an external recording solution containing (in mM): 125 NaCl, 2.5 KCl, 1.5 $CaCl_2$, 1.5 $MgCl_2$, 1.25 $NaH_2PO_4$, 24 $NaHCO_3$, 25 glucose and 0.5 ascorbic acid, and maintained at room temperature for up to 6 hr.

### Electrophysiology

Whole-cell patch-clamp recordings were performed from BCs, located in the inner third of the molecular layer of acute parasagittal slices (200 μm thick) of cerebellar vermis. Unless otherwise stated (*Figures 3* and *4*), 10 μM of SR-95531 was added to the ACSF to block GABA$_A$ receptors. Whole-cell patch recordings were performed using a Multiclamp 700B amplifier (Molecular Devices) at elevated temperatures (~32°C). We used fire-polished thick-walled glass patch electrodes (tip resistances of 4–6 MΩ). Patch pipettes were backfilled with the following internal solution (in mM): 115 $KMeSO_3$, 40 HEPES, 1 EGTA, 6 NaOH, 4.5$MgCl_2$, 0.49 $CaCl_2$, 0.3 NaGTP, 4 NaATP, 1 $K_2$-phospocreatine and 0.02 Alexa-594 or 0.04 Alexa-488 (adjusted to 300–305 mOsm, pH = 7.3, referred to as K$^+$-based internal solution); 115 $CsMeSO_3$, 40 HEPES, 1 EGTA, 6 NaOH, 4.5$MgCl_2$, 0.49 $CaCl_2$, 0.3 NaGTP, 4 NaATP, 1 Tris-phospocreatine and 0.02 Alexa-594 (adjusted to 300–305 mOsm, pH = 7.3, referred to as Cs$^+$-based internal solution). Series resistance was 14.0 ± 5.8 (mean ± SD, estimated from n = 160 cells) and always under 30 MΩ. All values of membrane potential were corrected for liquid junction potential, estimated to be −8 mV for K$^+$-based internal solution and −11 mV for Cs$^+$-based internal solutions (*Figure 1*; *Abrahamsson et al., 2012*). Holding potentials in voltage clamp were −70 mV, unless otherwise indicated. For current-clamp recordings, a bias current was injected to maintain the membrane potential (holding potential) at −70 mV, unless otherwise indicated. Series resistance was compensated by balancing the bridge and compensating pipette capacitance. In experiments described in *Figures 1* and *5*, internal solutions were further complemented with 1 mM QX-314 to prevent AP generation after entry of PF-mediated EPSCs.

### Pharmacological compounds

D-AP5 (D-(-)−2-Amino-5-phosphopentanoic acid), SR 95531 (2−3-Carboxyprobyl)−3-amino-6(4-methoxyphenyl pyridazinium bromide), and QX-314 chloride were purchased from Abcam, Cambridge, UK. NBQX (2,3-Dioxo-6nitro-1,2,3,4-tetrahydrobenzo[f]quinoxaline-7-sulfonamide) was purchased from Tocris Bioscience, Bristol, UK. Alexa Fluor 488 and 594 were purchased from Life Technologies, Carlsbad, California, USA. Mefloquine hydrochloride and DMSO (Dimethyl sulfoxide) were bought from Sigma-Aldrich, St Louis, Missouri, USA.

### Data analysis and statistics

Electrophysiological signals were low-pass filtered at 10 kHz, digitized at 100 kHz using an analog-to-digital converter (NI USB 6259, National Instruments), then acquired and analyzed using the Neuromatic analysis package (*Rothman and Silver, 2018*; www.neuromatic.thinkrandom.com) written within the Igor Pro environment (Wavemetrics, Portland, Oregon, USA). Peak amplitudes of synaptic responses recorded in voltage clamp or current clamp were measured as the difference between a baseline amplitude, estimated from 10 ms immediately preceding the stimulation artifact, and the mean amplitude over a 200 μs time window centered around the peak of the mean response. Due to the long-lasting nature of the after-hyperpolarization component of spikelets, the baseline subtracted peak amplitude of the inhibitory component was averaged over a time window of 3 ms, centered around the peak amplitude of the mean response. For the experiments in which we performed direct stimulation of neighboring MLIs, the lack of a synaptic delay required subtraction of the stimulus artifact, which we performed using a function in Neuromatic. In brief, the decaying inward component of the artifact was subtracted by first fitting an extrapolated double exponential function

between the time of the peak of the averaged inward component of the artifact to just before the spikelet. The fitted function was then subtracted from all the traces. The remaining mean artifact, comprising the outward component and rising inward component of the mean artifact, were subtracted directly from all traces.

All traces displayed are averages of 20 to 30 sweeps unless otherwise noted. They were filtered off-line using a binomial smoothing equivalent to a 4 kHz Bessel filter. For estimation of CCs we averaged 50 sweeps. Cells with high frequency (>5 Hz) spontaneous EPSCs or spikelet activity were not analyzed.

Measurement of distances between extracellular stimulation pipette locations and somata (*Figure 1*) or between the somata of two neighboring cells (*Figure 4—figure supplement 1*) was performed on 2PLSM maximal intensity projections and Dodt images in ImageJ with a freehand line tool, compensated for differences in z-offsets.

Data are expressed as averages ± SEM unless otherwise indicated. Statistical tests were performed using a non-parametric Mann-Whitney two-sample rank test routine for unpaired comparisons, or Wilcoxon matched-pairs signed rank test for paired comparisons unless otherwise stated. Kruskal-Wallis tests, followed by Dunn's multiple comparison tests, were employed to compare multiple groups within a single experiment. Linear correlations were determined by Pearson correlation coefficient. Statistical significance was defined as $p < 0.05$. All statistical tests were two-tailed, and performed in GraphPad Prism 6.

## Parallel fiber-mediated synaptic responses

Extracellular stimulation via patch electrodes filled with ACSF was performed using 50 μs voltage pulses using a constant voltage stimulator (Digitimer Ltd, Letchworth Garden City, UK). In the experiments shown in *Figure 1*, the external ACSF contained 2 mM $CaCl_2$, 1 mM $MgCl_2$, 10 μM gabazine (SR-95531) and 50 μM D-AP5 to block $GABA_AR$ and NMDAR, respectively. In our attempt to block the PF-mediated outward current (*Figure 1*), we also added in the ACSF either 20 μM mefloquine + 0.1% DMSO, or simply 0.1% DMSO. In all other experiments, we used 1.5 mM $CaCl_2$ and 1.5 mM $MgCl_2$, which are closer to physiological values (*Silver and Erecińska, 1990*; *Bouvier et al., 2018*).

Stable EPSC responses from parallel fibers (on-beam stimulation) were obtained by positioning a monopolar glass electrode (similar to patch electrodes, but filled with ACSF, and visualized by Dodt contrast) on top of the slices, above the dendrites of the patched MLIs (visualized by 2PLSM fluorescence of Alexa 594, introduced through the patch pipette). Stimulus intensities were set to 5–10 V above threshold for detecting an EPSC (*Abrahamsson et al., 2012*), referred to as low stimulation. To trigger spikelet responses (off-beam stimulation and direct stimulation), the stimulation pipette was positioned intentionally outside of the dendritic tree, and the relationship between electrical stimulation intensity and spikelet recruitment was systematically examined (*Figure 3*). The stimulation intensity was adjusted to between 20 and 80V in order to obtain detectable spikelet responses in at least 25–30 trials (see *Figures 3F–3J* and *8*). Outward currents were considered detectable if the amplitude distribution of peak outward currents for each trial was statistically larger than a prestimulus amplitude distribution within a reflected 3 ms time window around the baseline (same Δt from baseline as baseline region to the time of inward current peak). The distributions were compared with a Wilcoxon matched-pairs signed rank test.

## Transmitted light and fluorescence imaging

BC somata in the inner third of the molecular layer were identified and whole-cell patch clamped using infrared Dodt contrast (Luigs and Neumann, Ratingen, Germany) and a QIClick digital CCD camera (QImaging, Surrey, BC, Canada) mounted on an Ultima multiphoton microscopy system (Prairie Technologies, Middleton, Wisconsin, USA) based on an Olympus BX61W1 microscope, equipped with a water-immersion objective (60X, 1.1 NA, Olympus Optical, Tokyo, Japan). Two-photon excitation of Alexa-488 and Alexa-594 was performed at 780 nm for pairs of BCs. When Alexa-594 alone was used (single-cell patch-clamp), it was excited at 840 nm. A transmitted light PMT was mounted after the Dodt tube in order to acquire a pulsed IR laser-illuminated contrast image simultaneously with the 2P-LSM image. When the whole morphology of one cell could not be captured within a single stack of images in which the focal plane was varied (Z-image stacks; *e.g.*,

*Figure 1A*), multiple Z-image stacks were recorded and pairwise stitched in ImageJ (*Preibisch et al., 2009*).

## Detecting electrical and/or chemical synapses in paired recordings

The presence of an electrical synapse was assessed by analyzing CCs or spikelet transmission. Unidirectional CCs were calculated by dividing the average hyperpolarization in the non-injected cell (100 ms average) by the hyperpolarization value (100 ms average) recorded in the injected cell. Bidirectional CCs for a given BC-BC pair were calculated from the average of the two unidirectional CCs. Spikelet transmission was assessed by detecting postsynaptic inward currents in response to presynaptic AP firing since outward currents could arise from either spikelets and/or GABAergic currents. Inward current amplitudes were estimated from each sweep, following the baseline procedure described above, by averaging data points over a 200 µs window and occurring within a 2 ms window starting just after the time of the peak of the presynaptic AP. Inward current amplitudes were considered detectable if the amplitude distribution of peak inward currents was larger than a prestimulus amplitude distribution (Wilcoxon matched-pairs signed rank test, $p < 0.05$), measured from a reflected time window identical to that used for peak amplitudes. If inward currents were not detected, then outward currents in the postsynaptic responses were used to infer the presence of a GABAergic synapse. However, when an electrical synapse was detected, the outward current in postsynaptic responses could be caused by a mix of electrical and chemical inputs. In that case, peak amplitudes of postsynaptic outward currents were compared at different holding membrane potentials of the postsynaptic cell (to change the electromotive force for GABAergic inputs), and/or before and after application of gabazine. The presence of a GABAergic synapse was inferred when the peak amplitude distributions of outward currents in these different conditions were significantly different from each other. When an electrical connection was present, bias current in the presynaptic cell was manually adjusted, to compensate for passive flow of different holding currents from the postsynaptic cell.

To calculate the error in our estimate of the frequency of observing electrical and chemical synapses in paired recordings, we first computed the observed frequencies from our data set (electrical: $f_e = n_e/n_{pairs}$ and for chemical: $f_c = n_c/n_{unidirectional}$; where $n_e$ and $n_c$ are the number of bidirectional electrical and unidirectional chemical synapses, respectively). We assumed the frequency distributions could be described by binomial statistics, and therefore calculated a standard deviation ($SD = f*(1-f)$) and a standard error ($SEM = SD/\sqrt{n}$). We performed similar error calculations for all observation frequencies (e.g. *Figure 1*).

## Acknowledgements

This work is supported by the Centre National de la Recherche Scientifique, Fondation pour la Recherche Médicale and the Agence Nationale de la Recherche (ANR-2010-BLANC-1411, ANR-13-BSV4-0016, ANR-17-CE16-0019, ANR-17-CE16-0026 and ANR-18-CE16-0018), which were awarded to DAD. DAD's laboratory is a member of the BioPsy Laboratory of Excellence. We would like to thank Jeff Diamond for providing Cx36 KO animals. We also thank Jeff Diamond, Nelson Rebola, Angus Silver, Andreas Frick and Alexandra Cayco Gagic for useful comments on the manuscript. AH's PhD fellowship was funded by Sorbonne Université - Collège Doctoral ED3C and Fondation pour la Recherche Médicale (FDT-201805005603).

## Additional information

### Funding

| Funder | Grant reference number | Author |
|---|---|---|
| Centre National de la Recherche Scientifique | | David DiGregorio |
| Fondation pour la Recherche Médicale | | Andreas Hoehne Maureen H McFadden David DiGregorio |
| Agence Nationale de la Re- | ANR-2010-BLANC-1411 | David DiGregorio |

| | | |
|---|---|---|
| cherche | | |
| Agence Nationale de la Re-cherche | ANR-13-BSV4-0016 | David DiGregorio |
| Agence Nationale de la Re-cherche | ANR-17-CE16-0019 | David DiGregorio |
| Agence Nationale de la Re-cherche | ANR-17-CE16-0026 | David DiGregorio |
| Agence Nationale de la Re-cherche | ANR-18-CE16-0018 | David DiGregorio |

The funders had no role in study design, data collection and interpretation, or the decision to submit the work for publication.

### Author contributions

Andreas Hoehne, Conceptualization, Data curation, Formal analysis, Funding acquisition, Validation, Investigation, Methodology, Writing - original draft, Writing - review and editing; Maureen H McFadden, Data curation, Formal analysis, Writing - review and editing, MM performed wild-type recordings in animals more than 3-months old; David A DiGregorio, Conceptualization, Data curation, Supervision, Funding acquisition, Validation, Methodology, Writing - original draft, Project administration, Writing - review and editing

### Author ORCIDs

Maureen H McFadden (iD) https://orcid.org/0000-0001-9770-7069
David A DiGregorio (iD) https://orcid.org/0000-0002-6417-4566

### Ethics

Animal experimentation: All experiments were approved by the Ethics Committee #89 of Institut Pasteur (CETEA; protocol approval #DHA180006).

### Decision letter and Author response

Decision letter https://doi.org/10.7554/eLife.57344.sa1
Author response https://doi.org/10.7554/eLife.57344.sa2

## Additional files

### Supplementary files

• Transparent reporting form

### Data availability

All analyzed data used for this study are included in the manuscript and supporting files.

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
