## [Decision Letter]

**Acceptance summary:**

The authors describe a mechanism in the cerebellar cortex by which electrical coupling of inhibitory interneurons produces a specialized feedforward motif that enhances temporal summation at very short time intervals while shortening the window of integration by inhibiting temporal summation outside this brief window. The important conclusion of this work is that the well-known rapid feedforward inhibition of Purkinje cells driven by parallel fiber activation has both a chemical component, and, via the BC network, an electrical component.

**Decision letter after peer review:**

Thank you for submitting your article "Feed-forward recruitment of electrical synapses enhances synchronous spiking in the mouse cerebellar cortex" for consideration by *eLife*. Your article has been reviewed by Gary Westbrook as the Senior Editor, a Reviewing Editor, and three reviewers. The reviewers have opted to remain anonymous. The reviewers have discussed the reviews with one another and the Senior Editor has drafted this decision to help you prepare a revised submission.

Please note the changes in our revision policy in response to COVID-19 (https://elifesciences.org/articles/57162). Specifically, we are asking editors to accept without delay manuscripts, like yours, that they judge can stand as *eLife* papers without additional data, even if they feel that some additional experiments would make the manuscript stronger. Thus the revisions requested below primarily address clarity and presentation.

Summary:

The reviewers were interested in the topic and the proposed mechanism. However, we noted several issues in our discussion that need to be more carefully addressed in the text because the data in our opinion was not definitive. As noted above, *eLife* policy at this time is not to ask for additional experiments if the manuscript can be revised to state only those conclusions that are well supported by the existing data. We focused our discussion on the 5 issues below that we intend as a guide to your revision. Please also address other points of discussion or clarification raised in the original reviews that are included below.

Essential revisions:

1) Weak effects of KO/mefloquine. These effects were rather modest given that they are central to the proposed mechanism. The authors should discuss the possibility that ephaptic coupling could account for some component of the phenomenon, and the KO data need to be better integrated into the main text.

2) Comparison of GJs vs GABA in inhibiting the network. The reviewers were not convinced regarding the relative contribution of these components, which require a more balanced discussion.

3) Conclusions regarding resting membrane potential. Here again the reviewers were not convinced by the data presented. The authors should only make conclusions that are well supported by the data.

4) To better clarify the impact of your work, please point out similarities and differences to previous studies in the Discussion section.

5) The computation of connection probabilities needs to be better clarified.

We think these revisions can be made without increasing the length of the manuscript by eliminating redundancies in the text.

Reviewer #1:

In the manuscript "Feed-forward recruitment of electrical synapses enhances synchronous spiking in the mouse cerebellar cortex" Hohne and colleagues describe a mechanism in the cerebellar cortex by which electrical coupling of inhibitory interneurons produces a specialized feedforward motif that enhances temporal summation at very short time intervals while shortening the window of integration by inhibiting temporal summation outside this brief window. Together, these combined mechanisms are proposed to support a globalize and temporally precise inhibition of Purkinje neurons that could enhance synchrony thought to be important for information transmission downstream. The manuscript presents a progression of experiments describing the phenomenology and mechanism of the electrical coupling of basket cells, including evidence that outward currents observed in BCs are mediated in part by gap junctions; the spatial relationship of stimulation and the observation of electrical spikelet mediated outward currents; analog effects on electrical transmission; relative prevalence of electrical and chemical inhibition; and finally, the effect of electrical coupling on temporal summation of EPSPs. The data are thought provoking and of high quality and contribute to an active literature on MLI organization and physiology.

That said, the study has some shortcomings, which should be addressable with mainly text clarifications or additional analyses. I recognize the impossibility of collecting new data in the short term, thus where new data suggested below are not obtainable, interpretations should be tempered and narrowed.

1) The primary takeaway is that for independent PF beams, BC networks can detect coincident excitation that will generate a global precise inhibition of downstream Purkinje neurons. For this to be true, the temporal windowing contrast enhancement effect shown in Figure 6 needs to be robust to the presence of chemical inhibition. To my read, gabazine was present, leaving this uncertainty unresolved. Superimposed chemical and electrical inhibition could have predictable or unpredictable effects on the network level phenomenon reported here, raising the importance of empirical data in this light. For instance, Figure 3 shows that membrane potential of the presynaptic partner in an electrical pair affects electrical transmission. If effective GABAergic inhibition caused hyperpolarization of the presynaptic partner the magnitude of electrical inhibition would be expected to be reduced. Of course, this prediction may not pan out, but the main point is that it is difficult to predict how robust the phenomena reported here are to a full suite of intact circuit mechanisms.

2) The identification of gap junctions as mediating the outward currents measured is surprisingly weak. In Figure 1 they show that the outward current is unlikely mediated by unclamped K currents from direct stimulation, since it is robust to Cs internal and moreover the gap junctional antagonist mefloquine reduces the magnitude and frequency of observing these currents. However, a very important dataset is relegated to a supplementary figure that is in turn poorly integrated into the findings, specifically, experiments performed in Cx-36 knockouts shown in Figure 2. Although the title of the figure (S3) is 'Outward currents are smaller in SCs than BCs' the main text refers to the figure in the context of testing for the presence of outward currents in Cx-36 knockouts. The data are a bit of a mess with different age ranges used, precluding a clean interpretation. The data show that the amplitude of the outward currents recorded in Cx-36KO mice at P90-150 are about the amplitude as those recorded in wildtypes at P30-60 (the range used in the rest of the paper). There is a clear decrement in the amplitude of the putative electrical inhibition between wildtype and knockouts at a given age range, however, supporting the conclusion that this is gap-junction mediated. Given the extensive literature on ephaptic signaling in this part of the molecular layer, some caution is warranted in interpreting these as a purely jap-junctional mediated mechanism. Together, the mixing of experimental conditions and the persistence of the outward current in the knockouts seems to undermine one of the main takeaways of the study, sowing confusion and doubt. The authors should make some effort at better explaining and integrating this dataset into the main part of the study.

3) I have no reason to doubt the authors on the description of the inward and outward currents shown in Figure 1, or their sensitivity and insensitivity to blockade by receptor blockers, but the results as presented are very qualitative. Please report analyses in the text. For example, "Bath application of 10 μm .… NBQX eliminated both the EPSC and subsequent outward current (n = 6/6 experiments) should provide amplitude info +/- drug, as should the gabazine experiment before that.

4) Figure 3 data presents an interesting experiment testing how RMP of one or the other MLI in a coupled pair influences the spikelet polarity. They identify CC with paired recordings and see that the membrane potential of the presynaptic cell has a strong effect on the spikelet waveform but not vice versa. Some confusion in the exposition: The example in B shows bidirectional coupling but they then focus in unidirectional coupling. It is unclear if the cells used in D-F are unidirectionally or bidirectionally coupled or whether the holding potential manipulations of the 'pre and post' synaptic cells are filtered into the other. This overall raises the question of whether the correlations they show in H are based solely on the membrane potential of the presynaptic cell or on the differential membrane potentials between the two coupled cells. This could be clarified with edits.

5) For data shown in Figure 4, please report the number of cases where chemical inhibition was distinguished with pharmacology vs driving force. Also, I am missing how the probability distributions of chemical and electrical connections are compared and quantified. The results report that the probability of bidirectional transmission of spikelets was 0.52 +/- 0.06 (31/60 pairs). What does the error here represent? Similar with the GABAergic synapse: 0.40 +/- 0.05% (34/86 pairs). Finally, these probability distributions are compared with an unpaired t-test and shown to be not significantly different, but I am having difficulty following the mechanics of this analysis.

Reviewer #2:

In this study from the DiGrigorio lab, electrical synapses within the basket cell (BC) network were examined for their capacity to drive precisely timed activity, and thus inhibition of Purkinje cells. The most interesting conclusions are contained in Figure 5, which shows that narrow band synaptic activation of BCs can drive excitation of neighboring BCs with a temporal precision exceeding that of conventional EPSPs. The basis for this precision is that the excitation comes by way of spikelets and each spikelet is curtailed by the potent electrotonic spread of spike afterhyperpolarizations. An important conclusion then is that the well-known rapid feedforward inhibition of Purkinje cells driven by parallel fiber activation has both a chemical component, and, via the BC network, an electrical component. Thus, the authors coin the acronym eFFM, electrical feedforward modulation. This is good quality electrophysiology from a top lab, so I no major technical concerns. I do feel that apart from the above, the majority of the work is demonstrating rather incremental or expected things. For example, the generation of postsynaptic inhibition through the spikelet waveform is well described elsewhere. Also, I am not so convinced by the RMP estimate.

The use of the ratio of inward to outward current to guesstimate prejunctional resting membrane potential seems not very strong. This ratio is affected by other things besides RMP, for example: during the whole cell recording there can be washout of currents that underlie AHPs, there may be alterations in junctional conductance (which will filter the fast inward current more than the slow outward current). Indeed for the direct stim experiments, the slope of the graph in 5H is nearly twice that of the -60 mV graph. Do you really think the RMP is nearly -50 mV? If the prejunctional cell population is firing chronically, how would that impact this estimate?

Reviewer #3:

In this report, Hoehne et al., examine a novel type of feedforward inhibition mediated by an electrically coupled interneuron network in the cerebellum. They find that basket cells, a type of molecular layer interneuron, show curtailed synaptic excitation due to the offsetting effect of an after-hyperpolarization of a shared spikelet, generated in a neighboring electrically coupled basket cell. Interestingly, the depolarizing phase of the spikelet can cause increased firing in the postsynaptic cell. The combination of these opposing effects producing a type of contrast enhancement. Overall, this is an elegant, well-executed study. It is important for two reasons. First, the uniqueness of the underlying mechanism of electrical feedforward inhibition (FFI) whose influence differs from that mediated by chemical connections. Second, molecular layer inhibition is an important feature for organizing the output of the cerebellum, thus, contrast enhancement as described in this report could have a substantial effect on information coding in the cerebellum's circuits. The authors should consider revising their manuscript in several places to improve clarity and increase the impact of their work. These criticisms do not take away from overall enthusiasm.

1) The authors argue that electrical coupling is mediated by gap junctions but the effects of Cx36 knockout are very limited (e.g., 25% reduction in the frequency of observing the after-hyperpolarization of the coupled spikelet). The Marty group showed that MLI electrical coupling was absent in Cx36 KO mice so this is a very important caveat to the interpretation of their results. Did the authors consider the effects of Cx36 KO on temporal summation or on contrast enhancement? The authors should elaborate on the possibility of developmental compensation or alternative mechanisms that could explain this phenomenon in the Discussion section (ephaptic coupling?). Lastly, the presentation of the Cx36 data seems out of place in the Results section. Perhaps this was due to the lack of explanation in their methodology (e.g., 'coupled pairs' aren't well explained at first; only later in the manuscript was this described as recording from two basket cells at once).

2) The authors attempt to compare the prevalence of electrical vs chemical FFI is confusing. Why not simply state the rate of identifying electrically and chemically connected pairs? Then report the rate of spikelet coupling (which seems to be higher than the rate identified by long, hyperpolarizing pulses). Again, could spikelet coupling reflect a different mechanism? Last, is the prevalence of electrical vs chemical FFI best described by the rate of electrically and chemically connected pairs of basket cells?

3) Can the authors always be sure the presynaptic coupled neuron is a basket cell? It looks like the stimulating pipette is place throughout the molecular layer where stellate cells (and their neurites) also reside.

4) Could temperature-dependent effects on spike width and after-hyperpolarization duration contribute to differences with results from the Marty group (i.e., the prominent depolarization noted in their report)? This may also be true for the resting membrane potential in the presynaptic cell which the authors clearly show affects the polarity of influence.

5) EPSP half-widths in 5B and 5D seem to be different (responses appear much more prolonged in 5B). Does this mean that many responses in 5D in the LS condition have coupled spikelets (perhaps too small to resolve as outward currents but still large enough to effect EPSP duration)? How does this affect the interpretation of their results?

---

## [Author Response]

Essential revisions:1) Weak effects of KO/mefloquine. These effects were rather modest given that they are central to the proposed mechanism. The authors should discuss the possibility that ephaptic coupling could account for some component of the phenomenon, and the KO data need to be better integrated into the main text.

We recognize that vague wording and presentation of critical control experiments supporting the mechanism of GABAergic-independent feed-forward inhibition may have obfuscated reader. To correct this and other major concerns, we have added new main figures, performed additional analysis, clarified the description of results, and considered alternative hypotheses for our findings in the Discussion section.

1) We now emphasize that the 70% block by mefloquine is consistent with a gap-junction mediated mechanism, as we used a sub-saturating concentration in order to prevent significant off-target effects.

2) We now show that the coupling coefficient, a proxy of gap junction coupling, is positively correlated with the size of spikelets (*i.e.* peak amplitude of inward and outward spikelet currents), suggesting that spikelets are mediated by gap junctions.

3) Regarding the less than complete knock down of electrical coupling in the Cx36 KO mice, we now: a) highlight that there is still a 3-fold reduction in the outward current when compared to age-matched controls, b) discuss literature supporting the possibility that expression of Cx45 might compensate for Cx36 knock down, and c) discuss differences between our study and those of previous publications. In particular, we discuss the possibility that the use of older animals (>P90) vs. the 3 weeks old animals used in previous studies could allow for compensatory mechanisms to be expressed.

4) We now included a thorough discussion of the potential for spikelet transmission by ephaptic coupling. In brief, the large amplitude of BC spikelet currents is not compatible with the direct ephaptic currents generated between BC terminals and PC axon initial segments (Blot et al., 2016). Nor are they compatible with the sodium channel current evoked by ephaptic coupling between PC axons, since our spikelets are insensitive to internal QX314 and alterations in postsynaptic membrane potentials (Han and Regehr, 2018).

2) Comparison of GJs vs GABA in inhibiting the network. The reviewers were not convinced regarding the relative contribution of these components, which require a more balanced discussion.

Indeed, it is difficult to assess the relative contribution of chemical and electrical synapses to inhibitory transmission given the current data set. We have now added a paragraph in the Discussion section to highlight the limitations of our ability to assign a specific physiological role. Nevertheless, the observed higher frequency of electrical synapses implies a prominent and important mechanism. In the first submission there was an error in our spreadsheet, and upon correction, the higher frequency of electrical synapses is actually statistically significant. Moreover, we also now show that saturating concentrations of GABA_A_R antagonists did not statistically alter AP-evoked outward currents in paired BC recordings in 13 out of 21 cases. When Gabazine did have an effect, it blocked by only 50%.

We feel it is also important to highlight that spikelet-mediated inhibition, unlike chemical inhibition, is also coupled to a rapid initial depolarizing component that we show increases spike probability in addition to narrowing the time window for spiking, and they can propagate across multiple cells, ensuring entrainment of the network not otherwise possible with chemical inhibition.

Finally, we now mention in the Discussion section that future in vivo experiments are necessary to assess the physiological contribution of feed-forward spikelet transmission.

3) Conclusions regarding resting membrane potential. Here again the reviewers were not convinced by the data presented. The authors should only make conclusions that are well supported by the data.

We have carefully rewritten the Results sections so as not to overstate our conclusions. But the well-known voltage dependence of the relative hyperpolarization component, we feel, is a good indicator that MLIs have a depolarized resting potential. This is also consistent with the high propensity to evoke spikes in neighboring cells. But, independent of the resting membrane potential, the prominent biphasic spikelet is a recording of an unperturbed cell and likely to represent physiologically relevant characteristics.

4) To better clarify the impact of your work, please point out similarities and differences to previous studies in the Discussion section.

In the Discussion section, we now specifically comment on differences between our conclusions and those described previously. In brief, beyond the novel demonstration that spikelets are recruited in a feed-forward manner, we also show that the biphasic spikelet transmission is likely more physiologically relevant. We hypothesize that previous work in the same cell type did not observe the hyperpolarization component due to their hyperpolarized holding membrane potentials.

5) The computation of connection probabilities needs to be better clarified.

We have re-written the text and provided a new table to clarify the frequency of different criteria used to estimate connection probabilities of chemical and electrical connections between BCs.

We think these revisions can be made without increasing the length of the manuscript by eliminating redundancies in the text.

We have also made an effort to reduce redundancies to a minimum. However, to address all the comments of the reviewers, the Discussion section is indeed longer.

In summary, we are thankful for the reviewers’ comments as they have led to a greatly improved manuscript by encouraging us to revise analyses, calculations and presentation of the results. We have increased the number of main figures from 6 to 8, and hope that together with our additional changes we have made a more convincing argument that feed-forward recruitment of spikelet transmission if mediated by gap junctions and provides a powerful mechanism for synchronizing neural activity in ways not possible by chemical inhibition. We hope that the current manuscript is now suitable for publication in *eLife*.

Reviewer #1:In the manuscript "Feed-forward recruitment of electrical synapses enhances synchronous spiking in the mouse cerebellar cortex" Hohne and colleagues describe a mechanism in the cerebellar cortex by which electrical coupling of inhibitory interneurons produces a specialized feedforward motif that enhances temporal summation at very short time intervals while shortening the window of integration by inhibiting temporal summation outside this brief window. […] The data are thought provoking and of high quality and contribute to an active literature on MLI organization and physiology.That said, the study has some shortcomings, which should be addressable with mainly text clarifications or additional analyses. I recognize the impossibility of collecting new data in the short term, thus where new data suggested below are not obtainable, interpretations should be tempered and narrowed.

We thank the reviewer for the encouraging words and describe below how we have addressed the concerns.

1) The primary takeaway is that for independent PF beams, BC networks can detect coincident excitation that will generate a global precise inhibition of downstream Purkinje neurons. For this to be true, the temporal windowing contrast enhancement effect shown in Figure 6 needs to be robust to the presence of chemical inhibition. To my read, gabazine was present, leaving this uncertainty unresolved. Superimposed chemical and electrical inhibition could have predictable or unpredictable effects on the network level phenomenon reported here, raising the importance of empirical data in this light. For instance, Figure 3 shows that membrane potential of the presynaptic partner in an electrical pair affects electrical transmission. If effective GABAergic inhibition caused hyperpolarization of the presynaptic partner the magnitude of electrical inhibition would be expected to be reduced. Of course, this prediction may not pan out, but the main point is that it is difficult to predict how robust the phenomena reported here are to a full suite of intact circuit mechanisms.

The reviewer is correct in that the data presented do not differentially address the effect of chemical and electrical inhibition on feed-forward modulation of BC spiking. Indeed, a proper exploration of this would require experiments to be performed in a fully intact circuit (in vivo), where physiological GC activity might differentially recruit electrical FFI or chemical FFI. We have now pointed out these shortcomings of our study and indicate that future experiments would be required to dissect the true relative contribution. Even in vitro, the precise dissection is challenging due to the uncertainty in reversal potential for GABA_A_Rs.

In subsection “Electrical synapse-mediated spikelets dynamically modulate BC EPSP-spike coupling”, we added the following text:

“Here we did not examine the combined contribution of cFFI and eFFI, as its true contribution requires an accurate estimate of the GABAR reversal potential, which has not been performed in mature MLIs. Moreover, differential recruitment of cFFI and eFFI in MLIs under physiological spatial temporal PF activation will require future experiments in awake animals. Nevertheless, we found that BCs are more likely to transmit spikelets than chemical inhibition to their neighbors (58 versus 40 %, Figure 6), which could account for the narrowed excitatory responses of BCs versus SCs in vivo (Chu et al., 2012). Recent in vivo findings show that BC spikes have a stronger influence on PC spiking than SCs (Arlt and Häusser, 2020), implying that the specific timing of BCs will strongly influence PC output.”

We also highlighted the important difference between chemical inhibition and electrical synapses: namely the depolarizing – synchronizing component, and the electrical propagation within the BC network, which are likely to entrain a greater fraction of BCs than chemical inhibition alone.

In subsection “Electrical synapse-mediated spikelets dynamically modulate BC EPSP-spike coupling”, we added the following text:

“Nevertheless, we found that the transient excitatory drive of spikelets significantly increased the peak amplitude of the compound synaptic potential when both signals were coincident (within a 4-6 ms window). Thus, unlike cFFI, electrical feed-forward modulation (eFFM) can shorten the temporal window for EPSP-spike coupling, thus improving spike precision, while simultaneously enhancing spike probability (Figure 8).”

In subsection “Electrical connectivity in the cerebellar cortex is tuned for rapid output synchrony”, we added the following text:

“This high degree of electrical connectivity can provide a global network mechanism spanning > 100 µm in the sagittal plane (Rieubland et al., 2014) for generating a FF modulation of PC spiking patterns through synchronous inhibition (i.e., excitation of MLIs) followed by a rapid disinhibition (i.e., MLI self-inhibition). It should be noted that chemical inhibition alone does not propagate within the network, but spikelet-mediated inhibition does.”

We also would like to point out to the reviewer that it seems likely that the concurrence of both chemical inhibition and action potentials within the presynaptic cell would simply amplify the presynaptic hyperpolarization, which in turn is transmitted as inhibition to the electrically coupled receiving cell. Alternatively, if chemical inhibition is transmitted to both the pre and postsynaptic cell, then indeed the hyperpolarizing component of the transmitted spikelet would be occluded, but the receiving cell would still experience inhibition. Nevertheless, the most important difference of electrical (spikelet) and chemical inhibition, is that the spikelet inhibition will be coupled to a depolarizing component, which will be increased under hyperpolarized conditions of the presynaptic cell (new Figure 5), and will be very rapidly transmitted, indeed faster than the synaptic delay for inhibition. Thus, in all cases, the depolarizing component will be transmitted to the postsynaptic cell. In this light, the temporal contrast enhancement figure provides important and physiologically relevant intuition as to how brief BC synchrony can be generated, with or without additional chemical inhibition.

2) The identification of gap junctions as mediating the outward currents measured is surprisingly weak. In Figure 1 they show that the outward current is unlikely mediated by unclamped K currents from direct stimulation, since it is robust to Cs internal and moreover the gap junctional antagonist mefloquine reduces the magnitude and frequency of observing these currents. However, a very important dataset is relegated to a supplementary figure that is in turn poorly integrated into the findings, specifically, experiments performed in Cx-36 knockouts shown in Figure 2. Although the title of the figure is 'Outward currents are smaller in SCs than BCs' the main text refers to the figure in the context of testing for the presence of outward currents in Cx-36 knockouts.

We have now moved the results regarding the KO to the main Figure 2 and corrected the title.

The data are a bit of a mess with different age ranges used, precluding a clean interpretation. The data show that the amplitude of the outward currents recorded in Cx-36KO mice at P90-150 are about the amplitude as those recorded in wildtypes at P30-60 (the range used in the rest of the paper).

We thank the reviewer for pointing out the shortcomings in the presentation of our results, which perhaps contributed to an apparent underwhelming conclusion. Nevertheless, we are confident that gap junctions are the likely mechanism for the following reasons, which we now emphasized in the Results section and Discussion section.

1) Regarding the incomplete block of the outward currents by mefloquine: in order to minimize off-target effects, we used 20 mM mefloquine, which is expected to block gap junctions by 70% (Cruikshank et al., 2004). The outward current was reduced to 33%, consistent with an entirely gap junction-mediated mechanism.

We added the following text to subsection “PF-triggered outward current mediated by electrical synapses in cerebellar basket cells”: “In brain slices prepared from Cx36 KO mice, we found a 45 % reduction in the frequency of observing outward currents, and the amplitude of the remaining events was 3-fold smaller than age-matched recordings (4.2 ± 0.8 vs. 12.7 ± 1.8 pA).”

2) We apologize for the confusion resulting from showing Cx36 data in a Supplementary figure. To correct this, we moved the Cx36 KO data from a supplementary figure to a main figure (now Figure 2), and now only show age-matched comparisons, which highlight the 3-fold reduction in outward current in Cx36 mice. We previously misstated the Cx36 reduction as only two-fold, rather than three-fold.

3) We also now describe the strong correlation between coupling coefficient and spikelet amplitude in a main figure (new Figure 4). This result shows that the larger the coupling coefficient, the larger the spikelets. They also further support a gap junction origin of the spikelet transmission between BCs.

There is a clear decrement in the amplitude of the putative electrical inhibition between wildtype and knockouts at a given age range, however, supporting the conclusion that this is gap-junction mediated. Given the extensive literature on ephaptic signaling in this part of the molecular layer, some caution is warranted in interpreting these as a purely jap-junctional mediated mechanism.

Indeed, it is possible that the remaining inward and outward current in mefloquine or in Cx36 KOs could result from transmission of an action potential between closely juxtaposed dendrites or axons. Ephaptic transmission has been shown to involve different mechanisms.

In Blot et al., 2014, BC-PC ephaptic transmission is thought to occur between the axon terminal structures, pinceau, and the axon initial segment of PCs. Both the capacitive and K-currents generate a local field that sufficiently hyperpolarizes the AIS membrane potential and reduces the probability of PC spiking. Their simulations suggest that the ephaptic-mediated hyperpolarization is likely around ~ 1 µV at the somata, and thus undetectable. Thus, the large mV changes we observe between two BCs seem incompatible with such a coupling, as it would likely have to be between two axon terminals or crossings between BC dendrites.

In contrast, Han et al., 2018 measure ephaptic currents in PC-PC pairs that are 10's of pA, but are due to regenerative activation of local AIS sodium conductances in the post-PC, as they are blocked by intracellular perfusion of QX314 (sodium channel antagonist), and eliminated by postPC hyperpolarization to -70 mV (Figure 4). In the new Figure 5 we show that the amplitude of the spikelet is insensitive to postsynaptic manipulation in paired recordings of BC-BC pairs. Moreover, all the outward current experiments in Figure 1 were performed in the presence of QX-314. Kim et al. also nicely showed that the extracellular field was only generated within 20 µm of the high conductance density region of the AIS (Figure 5G), and not the soma or axon. Thus, the spikelets that we report, or even the 30% remaining unblocked current (in the presence of mefloquine or Cx36 KO), are unlikely mediated by this sodium current-dependent ephaptic mechanism.

Together, the mixing of experimental conditions and the persistence of the outward current in the knockouts seems to undermine one of the main takeaways of the study, sowing confusion and doubt. The authors should make some effort at better explaining and integrating this dataset into the main part of the study.

Again, we apologize if some of the explanation or logical flow of our arguments was not clear, but we believe that we have performed multiple experiments to support gap-junction mediated FFM. In addition to changes in the Results section and figures (see above), we also modified the discussion to clearly state the reasons for our conclusion, and the possibility of other interpretations (ephaptic). See Discussion section:

“Extracellular stimulation of PFs in parasagittal slices in the presence of gabazine produced outward currents of 10 pA in BCs and less than 3 pA in SCs (Figure 1 and Figure 2). We performed several experiments whose results were consistent with spikelet transmission through gap junctions as the origin of GABAergic-independent outward currents: (1) sub-saturating concentrations (20 µM) of the gap junction blocker mefloquine blocked the outward current by 70 %, as expected for a current entirely mediated by gap junctions (Cruikshank et al., 2004). (2) We observed a 3-fold reduction in peak amplitude of the outward current in Cx36 KO mice (Figure 2). The incomplete knockdown could be due to genetic compensation by Cx45, which is known to also be expressed in MLIs (Van Der Giessen et al., 2006). Nevertheless, we only observed one of five BC-BC pairs in Cx36 KO mice that showed bidirectional spikelets and a CC greater than 2 % (the average CC correlated with the observation of a spikelet, Figure 2A), as opposed to the nearly 60 % BC-BC pairs (Figure 6C). (3) To confirm that the outward currents were mediated by spikelets, we showed that stimulation of PFs outside the dendritic field of the recorded cell and direct stimulation of MLIs in NBQX both evoked currents with a rapid inward component and an outward component similar to that following EPSCs. (4) These extracellularly-evoked spikelets were very similar to those evoked by single APs propagating between electrically coupled BC pairs. And finally, (5) the strength of electrical coupling between two BCs, estimated from 400 ms step depolarizations, was positively correlated with spikelet amplitude (Figure 4), confirming that spikelets were mediated by gap junctions.

Electrical coupling between neurons has also been shown to be mediated without a gap junction or chemical synapse, and is referred to as ephaptic coupling. We cannot directly rule out that the remaining spikelet currents in mefloquine or in Cx36 KOs could result from transmission of an AP between closely juxtaposed dendrites or axons. BC-PC ephaptic transmission is thought to occur between the axon terminal of BCs (pinceau) and the axon initial segment of PCs. Both the capacitive and K-currents generate a local field that sufficiently hyperpolarizes the AIS membrane potential and reduces the probability of PC spiking (Blot and Barbour, 2014). In this study, simulations suggest that the ephaptic-mediated hyperpolarization is likely around ~ 1 µV at the somata, and thus unlikely to account for the large mV changes we observe between two BCs. In contrast, ephaptic currents in PC-PC pairs are 10's of pA, but are due to regenerative activation of local AIS sodium conductances in the post-PC, which is eliminated by post-PC hyperpolarization to -70 mV (Han et al., 2018). We showed that the synapse-evoked outward current was insensitive to 1 mM of the sodium channel blocker QX-314 (Figure 1, see Materials and methods section); and, in BC-BC paired recordings, the amplitude of the spikelet is insensitive to postsynaptic manipulation of membrane potential (Figure 5). Both results are inconsistent with a sodium dependent ephaptic mechanism.”

3) I have no reason to doubt the authors on the description of the inward and outward currents shown in Figure 1, or their sensitivity and insensitivity to blockade by receptor blockers, but the results as presented are very qualitative. Please report analyses in the text. For example, "Bath application of 10 μm .… NBQX eliminated both the EPSC and subsequent outward current (n = 6/6 experiments) should provide amplitude info +/- drug, as should the gabazine experiment before that.

We have added a summary panel (Figure 1E) describing the NBQX blockade of the outward current. See added text in subsection “PF-triggered outward current mediated by electrical synapses in cerebellar basket cells”: “Bath application of 10 µM of the AMPA receptor (AMPAR) antagonist, NBQX, eliminated both the EPSC and the subsequent outward current to below detectable levels (Figure 1E, n = 5/5 experiments).”

Unfortunately, we did not characterize PF-evoked outward currents with and without gabazine, however, we did examine in some BC-BC pairs the effect of gabazine on the outward currents (see new Figure 4—figure supplement 1). We found that, in 13 out of 21 unidirectional connections from BC-BC pairs, the GABA_A_R antagonist did not alter detectably the outward current amplitude, and, in the remaining connections, the fractional block by 10 µM gabazine was ~50% on average (new Figure 4—figure supplement 1B). The following text was added to the Results section:

“If we analyzed the gabazine only experiments, we found that in 13 out of 21 unidirectional connections, the GABA_A_R antagonist did not alter the outward current amplitude. In the remaining connections the fractional block by 10 µM gabazine was ~50 % on average (Figure 4—figure supplement 1B).”

4) Figure 3 data presents an interesting experiment testing how RMP of one or the other MLI in a coupled pair influences the spikelet polarity. They identify CC with paired recordings and see that the membrane potential of the presynaptic cell has a strong effect on the spikelet waveform but not vice versa. Some confusion in the exposition: The example in B shows bidirectional coupling but they then focus in unidirectional coupling. It is unclear if the cells used in D-F are unidirectionally or bidirectionally coupled or whether the holding potential manipulations of the 'pre and post' synaptic cells are filtered into the other. This overall raises the question of whether the correlations they show in H are based solely on the membrane potential of the presynaptic cell or on the differential membrane potentials between the two coupled cells. This could be clarified with edits.

We have clarified in the text and in the methods that bidirectional coupling coefficients (CCs) are values that represent the average of the two unidirectional CCs. Unidirectional CCs are calculated from the ratio of the membrane potential in the receiving cell over the hyperpolarization amplitude of the cell in which the current injection was delivered.

Because current injections were made to maintain resting potentials in current clamp, and the voltage clamp circuit ensured a constant membrane potential, the resting membrane potentials were essentially decoupled, allowing for an accurate determination of the membrane potential dependence on the spikelet shape, without influencing the membrane potential of the electrically connected cell. In Figure 5E and F the receiving cell is held at -70mV, which might certainly influence the resting membrane potential of electrically-coupled neighbors.

5) For data shown in Figure 4, please report the number of cases where chemical inhibition was distinguished with pharmacology vs driving force.

This is a good suggestion. We have now added a supplementary table indicating the number of unidirectional connections tested for each method (Figure 4—figure supplement 1B).

Also, I am missing how the probability distributions of chemical and electrical connections are compared and quantified. The results report that the probability of bidirectional transmission of spikelets was 0.52 +/- 0.06 (31/60 pairs). What does the error here represent? Similar with the GABAergic synapse: 0.40 +/- 0.05% (34/86 pairs). Finally, these probability distributions are compared with an unpaired t-test and shown to be not significantly different, but I am having difficulty following the mechanics of this analysis.

Perhaps this was not well explained. The standard error we show is calculated from the standard deviation of a binomial distribution centered on the observed frequency of each connection type. Once having the SEM, we could perform a simple unpaired two-tailed t-test. This was originally briefly described in the figure legend, but now also described in the Materials and methods section: “To calculate the error in our estimate of the frequency of observing electrical and chemical synapses in paired recordings, we first computed the observed frequencies from our data set (electrical: f_e_ = n_e_/n_pairs_ and for chemical: f_c_ = n_c_/n_unidirectional connections_). We assumed the frequency distributions could be described by binomial statistics, and therefore calculated a standard deviation (SD = f*(1-f)) and a standard error (SEM = SD/<inline-graphic mime-subtype="png" mimetype="image" xlink:href="media/image1.png" />n). We performed similar error calculations for all observation frequencies (e.g. Figure 1).”

Reviewer #2:[…] The use of the ratio of inward to outward current to guesstimate prejunctional resting membrane potential seems not very strong. This ratio is affected by other things besides RMP, for example: during the whole cell recording there can be washout of currents that underlie AHPs, there may be alterations in junctional conductance (which will filter the fast inward current more than the slow outward current).

Indeed, it is possible that washout might alter the voltage-dependence of the AHP, and thus shift our calibration. The most likely candidate would be washout of calcium buffers that affect calciumdependent potassium conductances. However, we have included 1 mM EGTA in the pipette which approximates the concentration of the slow on-rate calcium binding protein parvalbumin. Nevertheless, the spikelets initiated from unperturbed cells have a large outward component consistent with a large presynaptic AHP, which is associated with APs initiated from resting membrane potentials much more depolarized than potassium current reversal potentials. Moreover, the fact that we reliably stimulate synapse-evoked spikes also suggests a depolarized resting membrane potential.

We now refer to the caveats of our membrane potential estimation in the discussion, and emphasize the bipolarity of the spikelets rather than the absolute value membrane potential, subsection “Electrical synapse-mediated spikelets dynamically modulate BC EPSP-spike coupling”:

“Comparison of spikelets evoked in an unperturbed, electrically coupled neighbor to those evoked in paired recordings with presynaptic membrane potential manipulations is consistent with a depolarized BC resting membrane potential (> -60 mV; assuming voltage-dependence of AP shape is equal between unperturbed and dialyzed recordings).”

Indeed for the direct stim experiments, the slope of the graph in 5H is nearly twice that of the -60 mV graph. Do you really think the RMP is nearly -50 mV?

We also find this result to be striking. We now state that our data are consistent with a resting membrane potential more depolarized than -60 mV, provided that the voltage dependence of the spikelet shape is similar for whole-cell dialyzed and unperturbed cells, although previous studies also estimated resting membrane potentials to be more depolarized than -60 mV (Chavas and Marty, 2003).

We have added the following sentence to the Results section:

“Provided that the voltage-dependence of the shape of the presynaptic AP is similar in patchdialyzed and unperturbed cells, these data indicate that unpatched MLIs have a depolarized resting membrane potential of at least -60 mV (i.e. BCs could be more depolarized).”

If the prejunctional cell population is firing chronically, how would that impact this estimate?

Because we could perform stimulus aligned averages, the background spikelet activity did not generally affect our experiments. Indeed, there were one or two cells out of the over 100 recorded cells that were not analyzed due to high background spikelet activity. We have added this criterion to the Materials and methods section.

Reviewer #3:1) The authors argue that electrical coupling is mediated by gap junctions but the effects of Cx36 knockout are very limited (e.g., 25% reduction in the frequency of observing the after-hyperpolarization of the coupled spikelet). The Marty group showed that MLI electrical coupling was absent in Cx36 KO mice so this is a very important caveat to the interpretation of their results.

Indeed, we were disappointed that the Cx36 KO mice did not completely eliminate spikelet transmission, although we did observe a 3-fold reduction of currents in age-matched controls. This was incorrectly described in the first manuscript as 2-fold. We also made several changes to figures and text to justify our conclusion despite the remaining currents. Please see response to comment number #2 of reviewer 1. In short, it is possible that because our animals are 3-4 months old, perhaps the remaining currents are mediated by another gap junction protein Cx45, also shown to be expressed in SCs.

Did the authors consider the effects of Cx36 KO on temporal summation or on contrast enhancement?

We agree that this would be an ideal experiment. But because Cx36 did not eliminate electrical coupling, instead we compared temporal summation at sites with and without eFFM. We felt that under the circumstances this would be a cleaner approach. Moreover, we could select dendritic locations in which PF stimulation evoked EPSPs that did not evoke a spikelet, thus allowing us to precisely probe the effect of spikelets evoked at different times within the EPSP, which itself does

The authors should elaborate on the possibility of developmental compensation or alternative mechanisms that could explain this phenomenon in the Discussion (ephaptic coupling?).

Indeed, Cx45 is expressed in MLIs and might underlie residual or compensatory spikelet transmission. Please see response to comment 2, reviewer 1 for full description of how we address this issue now in the manuscript, including the possibility of ephaptic coupling.

Lastly, the presentation of the Cx36 data seems out of place in the Results. Perhaps this was due to the lack of explanation in their methodology (e.g., 'coupled pairs' aren't well explained at first; only later in the manuscript was this described as recording from two basket cells at once).

This is a good suggestion, we now show the Cx36 data in the main Figure 2 and more clearly introduce BC-BC paired recordings in the Results section:

“In order to confirm that extracellular stimulation indeed evoked a presynaptic AP which could then propagate across electrical synapses formed by gap junctions to produce a spikelet in the postsynaptic cell, we performed paired BC-BC recordings. Hyperpolarizing current pulses were used to confirm the presence and the strength of coupling coefficients (CCs; see Materials and methods section).”

We also changed “coupled pairs” to “electrically coupled cells.”

2) The authors attempt to compare the prevalence of electrical vs chemical FFI is confusing. Why not simply state the rate of identifying electrically and chemically connected pairs?

We felt the use of “prevalence” was accurate, since it is often used to refer to the percentage of a population that is affected by a disease, and inherently also a frequency. But instead we replaced it with the words “more often” or “frequency” to avoid any confusion.

Then report the rate of spikelet coupling (which seems to be higher than the rate identified by long, hyperpolarizing pulses).

Perhaps our text was confusing, but the frequency of observing spikelets in BC-BC pairs is less than the detection of electrical coupling. We think this is for two reasons: (1) transient spikelet propagation through gap junctions are filtered more than the steady-state current components of long step depolarizations, and (2) some BC-BC pairs may not be directly connected, but connected through a third MLI. Nevertheless, we have added new panels to the new Figure 4 (D-F), which show a positive correlation between spikelet amplitude and coupling strength, as would be expected from being mediated by gap junctions.

Again, could spikelet coupling reflect a different mechanism?

Please see our responses to reviewer 1 for a thorough discussion of this issue.

Last, is the prevalence of electrical vs chemical FFI best described by the rate of electrically and chemically connected pairs of basket cells?

We no longer use the word “prevalence,” but either use “more often” or “frequency.”

3) Can the authors always be sure the presynaptic coupled neuron is a basket cell? It looks like the stimulating pipette is place throughout the molecular layer where stellate cells (and their neurites) also reside.

Indeed, the presynaptic cell could be a SC or a BC, but Rieubland show that electrical coupling of putative BC-SC paired recordings occurs at a much lower frequency (13% versus 61% putative BC-BC pairs) and have lower CC. We therefore refer to extracellularly evoked spikelets as coming from MLIs more generally rather than BCs. However, we do mention that more often than not, the spikelets are mediated by neighboring basket cells, based on the prominent electrical coupling and low probability of coupling between BCs and SCs (Rieubland et al., 2015).

In the Discussion section we updated the text:

“Our results indicate that low stimulation intensity of small PF beams leads to the recruitment of APs in electrically coupled MLIs that generate spikelets in the postsynaptic BCs (Figure 1, Figure 2, Figure 3), but more rarely and less prominently in SCs (Figure 2). Because of the low frequency (13 %) of BC-SC electrical coupling (Rieubland et al., 2014), we propose that eFFI is mostly provided by other BCs.”

4) Could temperature-dependent effects on spike width and after-hyperpolarization duration contribute to differences with results from the Marty group (i.e., the prominent depolarization noted in their report)? This may also be true for the resting membrane potential in the presynaptic cell which the authors clearly show affects the polarity of influence.

Alcami, 2018 show that there were no differences in spikelet shape if experiments are performed at 21 or 34 °C, so a temperature-dependence seems unlikely to account for the difference. However, as suggested by the reviewer, the most likely difference is due to the hyperpolarized holding potential (-70 mV in some cases), which we predict should have a small AHP. Interestingly, when they recorded a spikelet, the prominent outward current was reminiscent of our recordings (Alcami et al. 2013). Finally, we cannot rule out the possibility that in P11-13 rats, the AHP is less prominent. We have added the following text to the Discussion section.

“Comparison of spikelets evoked in an unperturbed, electrically coupled neighbor to those evoked in paired recordings with presynaptic membrane potential manipulations is consistent with a depolarized BC resting membrane potential (> -60 mV; assuming voltage-dependence of AP shape is equal between unperturbed and dialyzed recordings). We postulate that previous recordings showing mostly depolarizing spikelets in cerebellar MLIs were due to either hyperpolarized holding potentials (Alcami, 2018) or an intrinsic difference in the amplitude of the AHP recorded from brain

5) EPSP half-widths in 5B and 5D seem to be different (responses appear much more prolonged in 5B). Does this mean that many responses in 5D in the LS condition have coupled spikelets (perhaps too small to resolve as outward currents but still large enough to effect EPSP duration)? How does this affect the interpretation of their results?

The large distribution in EPSP decays is perhaps due to cable filtering when performing synaptic stimulation in different parts of the dendrite, similar to what we previously showed for stellate cells (Abrahamson et al., 2012.) However, it is also possible that BCs have different numbers of electrically coupled neighbors, which could decrease their input resistance (Alcami et al., 2013) and thus accelerate the decay of the EPSPs.

It seems unlikely that EPSCs without outward currents could have significantly altered half-widths, but we cannot strictly rule it out.

We have added the following text to the Results section:

“The variability in the EPSC decay in response to low stimulation could be due to the variability in cable filtering when eliciting synapse stimulation in different locations within the dendrite, or the variability in the number of electrically coupled cells, which have been shown to decrease input resistance (Alcami and Marty, 2013). But we cannot rule out that a small spikelet may accelerate some EPSPs without showing an outward current. Nevertheless, recruiting an additional spikelet, on average, accelerated EPSP decays”